# Auxiliary subunits reshape structural asymmetry and functional plasticity in heterotetrameric GluA1/A2 AMPA receptor core

Laura Y. Yen[1,2], Thomas P. Newton [1,3], Maria V. Yelshanskaya[1], Muhammed Aktolun [4], Shanti Pal Gangwar [1], Rasmus P. Clausen[5], Maria G. Kurnikova [4] & Alexander I. Sobolevsky [1] ✉

AMPA-subtype ionotropic glutamate receptors (AMPARs) mediate the fast component of excitatory neurotransmission. They govern synaptic plasticity that underlies learning and memory, while their dysregulation is implicated in numerous neurological disorders. The functional diversity of AMPARs arises from variations in their subunit composition and also their association with auxiliary subunits. While multiple structures of homomeric AMPARs have been reported, structural information for the heteromeric core – particularly in the absence of auxiliary subunits, which would serve as a functional and structural baseline – has been limited. Here, we report cryo-electron microscopy structures of GluA1/A2, the most abundant AMPAR di-heteromer in the brain, in the closed, open, and desensitized states. Using molecular dynamics (MD) simulations and cross-correlating structural and functional information, we find that auxiliary subunits increase the diameter of channel pore, which corresponds to larger conductance. Likewise, we find that recovery from desensitization slows with greater disruption of two-fold rotational symmetry of the ligand-binding domain dimer in the desensitized state. Both receptor activation and desensitization vary with the type and number of associated auxiliary proteins. These structures offer a foundation for uncovering how auxiliary subunits reshape structural asymmetry and functional plasticity in heterotetrameric AMPARs.

Excitatory neurotransmission in the mammalian central nervous system (CNS) is primarily mediated by ionotropic glutamate receptors (iGluRs), a family of tetrameric, ligand-gated, cation-permeable channels. Based on sequence homology and pharmacology, iGluRs are divided into four subclasses: α-amino-3-hydroxy-5-methyl-4- isoxazolepropionic acid receptors (AMPARs), kainate receptors (KARs), N-methyl-D-aspartate receptors (NMDARs), and delta receptors[1,2]. Among the iGluRs, AMPARs are distinguished by their exceptionally rapid gating kinetics. Glutamate (Glu) release from pre-synaptic vesicles into the synaptic cleft and subsequent binding to

[1]Department of Biochemistry and Molecular Biophysics, Columbia University, New York, NY, USA. [2]Cellular and Molecular Physiology and Biophysics Graduate Program, Columbia University Irving Medical Center, New York, NY, USA. [3]Cellular, Molecular, and Biomedical Studies Umbrella Program, Columbia University Irving Medical Center, New York, NY, USA. [4]Department of Chemistry, Carnegie Mellon University, Pittsburgh, PA, USA. [5]Department of Drug Design and Pharmacology, University of Copenhagen, Copenhagen, Denmark. ✉e-mail: as4005@cumc.columbia.edu

AMPARs activates the receptor, allowing flux of cations into the postsynaptic cell and subsequent depolarization of the postsynaptic membrane at millisecond timescales[3,4]. AMPARs are ubiquitously expressed in all mature excitatory postsynaptic neurons and play essential roles in synaptic plasticity, learning, and memory[5,6]. Dysfunction of AMPAR signaling has been implicated in various neurological disorders, including ischemic stroke, Alzheimer's disease, autism spectrum disorder, and epilepsy[7–9], positioning them as compelling therapeutic targets. Despite decades of pharmacological interest, efforts to develop AMPAR-targeted therapies have been hampered by poor compound solubility, renal toxicity, and CNS-related side effects, such as sedation and dizziness[10–13]. These challenges underscore the need for a more detailed molecular-level understanding of the mechanisms of AMPAR gating and pharmacology to inform rational therapeutic developments.

The AMPAR core assembles as a homo- or heterotetramer composed of subunits GluA1–A4, with distinct subunit combinations conferring unique functional properties. For instance, GluA2-containing AMPARs have generally low permeability to $Ca^{2+}$ due to RNA editing that substitutes glutamine (Q) for arginine (R) at position 586, the Q/R site, within the channel pore[14–16]. In contrast, GluA2-lacking AMPARs are $Ca^{2+}$-permeable and sensitive to block by polyamines, such as intracellular spermine and spermidine[17,18]. Furthermore, GluA1 and GluA4 are both crucial for long-term potentiation (LTP) and synaptic plasticity[19–22]. Moreover, AMPARs associate with auxiliary proteins and trafficking adaptors, which further refine their localization and function at distinct synaptic and extrasynaptic compartments[23,24]. In the brain, AMPARs primarily assemble as heteromers in complex with auxiliary subunits[25,26]. However, the majority of structural and functional studies of AMPAR-auxiliary subunit complexes have focused on assemblies with the homomeric AMPAR core[27–30], and most available structures of the homomeric core alone have suffered from both low resolution and absence of the critical re-entrant M2 loop of the transmembrane domain (TMD), which lines the pore in conjunction with helix M3 and forms the channel's selectivity filter[31–36]. Only one structure of a heteromeric AMPAR core—GluA2/A3—has been resolved without auxiliary proteins, but it adopts an unusual "O-shape" conformation, is of limited resolution, and remains in a non-conducting state with poorly resolved M2 loop[37], suggesting a common trend that auxiliary subunits serve as TMD stabilizers in structural studies. The absence of structural insight on intact heteromeric AMPAR cores, in the absence of auxiliary proteins, constrains our understanding of their baseline gating mechanisms and limits mechanistic interpretations of auxiliary subunit modulation.

GluA1/A2 is the most abundant heteromeric AMPAR core assembly in the brain[26,38]. GluA1 facilitates synaptic insertion during LTP, supporting dynamic AMPAR trafficking[39], while the $Ca^{2+}$-impermeable GluA2 subunit limits calcium influx, contributing to neuroprotection[40]. Together, these subunits coordinate the localization and functional tuning of AMPARs in the CNS. To date, the expression, purification, and structural characterization of the naked GluA1/A2 core has been unsuccessful, hindered by poor heterologous expression and low extraction efficiency—limitations that are typically alleviated by co-expression of the AMPAR core with auxiliary subunits[25,26,28,39–41]. Here, we use cryo-EM to elucidate the structure of the heteromeric GluA1/A2 AMPAR core, without auxiliary subunit binders, in its closed (resting), open (activated), and desensitized states. These structures reveal intrinsic asymmetries in the receptor core that suggest distinct allosteric gating features. Furthermore, we determined the structure of the activated hetero-octameric complex of the GluA1/A2 core bound to endogenous cornichons (CNIHs), auxiliary subunits whose paralogs CNIH2 and CNIH3 enhance AMPAR trafficking[42] and potentiate channel activity through the slowing of deactivation kinetics[38,43,44]. Our structural studies are supplemented by MD simulations, which characterize the channel conductance of the resolved protein conformations. Our

findings demonstrate a molecular and structural foundation for selective modulation by auxiliary subunits and provide evidence of divergences in mechanisms of desensitization, an essential process within receptor gating, based on the protein complex makeup. Altogether, our results expand the understanding of heteromeric AMPAR gating and outline the importance and framework for considering both core and auxiliary subunit composition for effective and specific therapeutic design to treat AMPAR-related neuropathies.

## Results

### Functional characterization of heteromeric GluA1/A2 AMPA receptors

To enable functional studies of heteromeric GluA1/A2, we modified GluA1 and GluA2 constructs based on previous structural studies[28,32]. For GluA2, we utilized the RNA-edited form with R586 at the Q/R site, mimicking naturally occurring GluA2. Both constructs were co-expressed for functional and structural studies (GluA1/A2$_{CryoEM}$). We tested the function of GluA1/A2$_{CryoEM}$ in comparison with wild-type GluA1/A2 (GluA1/A2$_{WT}$) using whole-cell patch-clamp recordings (Fig. 1). By measuring the time constant of the current decline after short, 2-ms application of agonist Glu at the saturating concentration of 3 mM, we determined the time constant of receptor deactivation, $\tau_{Deact}$ (Fig. 1a, green trace; Fig. 1d: 3.45 ± 0.40 ms, $n = 10$ for GluA1/A2$_{CryoEM}$; 3.34 ± 0.42 ms, $n = 7$ for GluA1/A2$_{WT}$). The time constant of desensitization, $\tau_{Des}$, was determined by single-exponential fit of current decline in response to the prolonged, 1 s application of Glu (Fig. 1a, blue trace; Fig. 1c: 7.64 ± 0.59 ms, $n = 17$ for GluA1/A2$_{CryoEM}$; 4.21 ± 0.48 ms, $n = 11$ for GluA1/A2$_{WT}$). The steady-state fraction of non-desensitized receptors ($I_{SS}/I_{max}$) was determined as a ratio of the amplitude of the steady-state current in response to prolonged application of Glu ($I_{SS}$) to the maximal current amplitude in response to the 1-s application of Glu in the presence of the positive allosteric modulator (PAM) cyclothiazide (CTZ) ($I_{max}$, Fig. 1a, orange trace; Fig. 1f: 0.032 ± 0.008 ms, $n = 12$ for GluA1/A2$_{CryoEM}$; 0.016 ± 0.005 ms, $n = 9$ for GluA1/A2$_{WT}$). The time constant of recovery from desensitization, ($\tau_{RecDes}$, Fig. 1e: 105 ± 17 ms, $n = 8$ for GluA1/A2$_{CryoEM}$; 104 ± 12 ms, $n = 8$ for GluA1/A2$_{WT}$), was estimated using the double-pulse protocol, in which a conditioning 80-ms pulse of Glu was followed by another one, after a period of time given to receptors for exiting desensitization[34] (Fig. 1b).

The four parameters had similar values between GluA1/A2$_{CryoEM}$ and GluA1/A2$_{WT}$ (Fig. 1c–f), although the time constant of desensitization for wild-type receptors was significantly (1.8-times) faster. While the slower rate of GluA1/A2$_{CryoEM}$ desensitization may be a consequence of the construct modification, such as shortening of the ATD–LBD linker, the lack of significant effect on the fraction of non-desensitized channels (Fig. 1f) and the rate of recovery from desensitization (Fig. 1e) suggest that such a small difference in the rate of desensitization is unlikely to cause a dramatic change in the structural mechanism of GluA1/A2$_{CryoEM}$ desensitization. Overall, the functional characteristics of GluA1/A2$_{CryoEM}$ were similar to GluA1/A2$_{WT}$, suggesting that the modified receptor, used for structural studies, is a fair representation of the native GluA1/A2 heteromer.

### Structures of heteromeric GluA1/A2 AMPA receptor

For structural studies, the heteromeric GluA1/A2 receptors were purified from HEK 293S GnTI- cells (see **Methods**) and subjected to single-particle cryo-EM in three different conditions, aiming to solve structures in the closed (in the presence of competitive antagonist ZK200775 (ZK)), open (in the presence of agonist Glu and PAM (R,R)−2b) and desensitized (in the presence of agonist quisqualate (Quis)) states (Fig. 2a–c and Supplementary Figs. 1–3). The three major populations of particles yielded 3.4-, 3.2-, and 4.5-Å resolution reconstructions of the core GluA1/A2 heterotetramer in the closed, open, and desensitized states, respectively (Fig. 2d–f and Supplementary

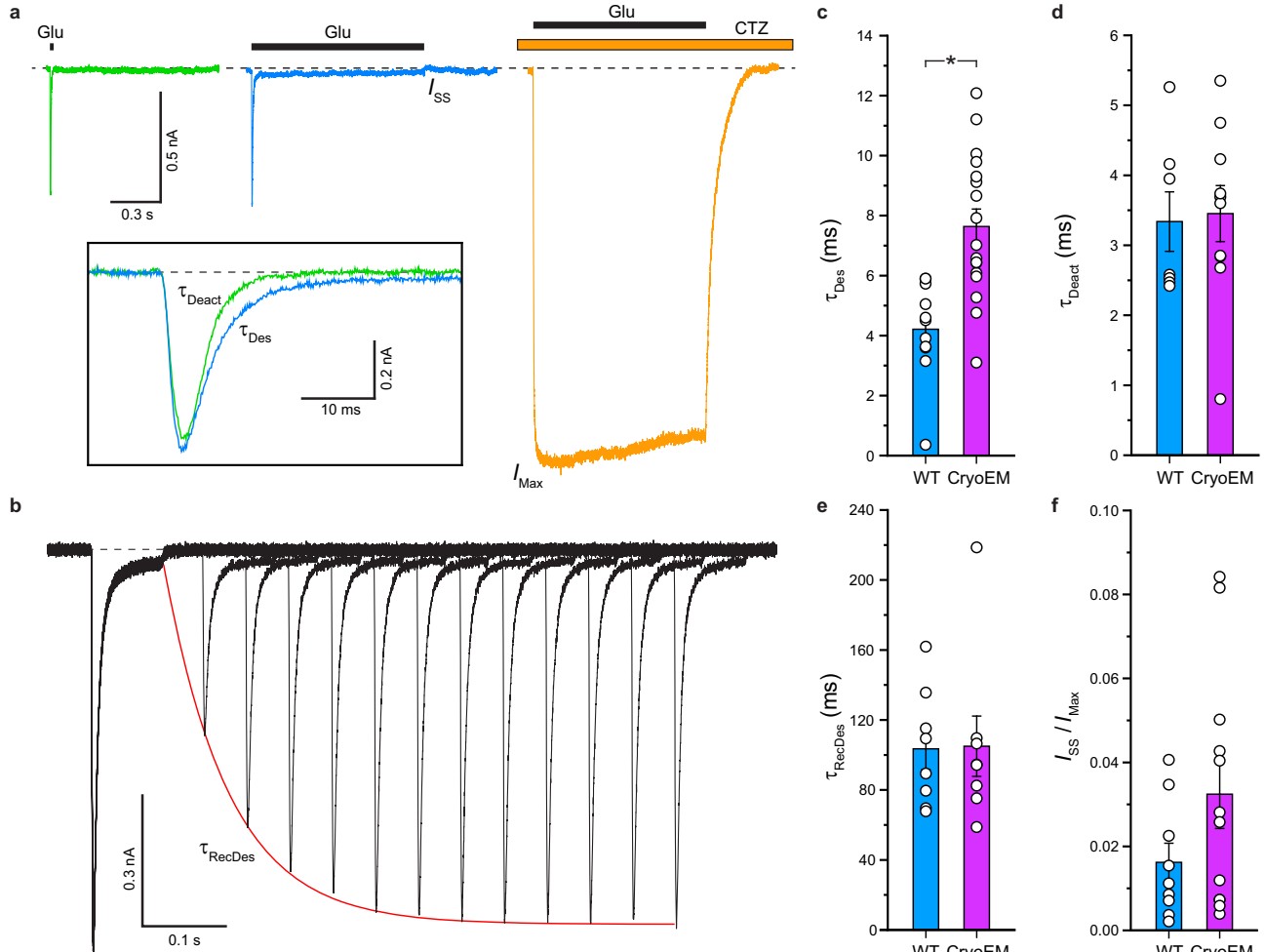

**Fig. 1 | Functional characterization of heteromeric GluA1/A2 receptors.**

**a** Representative whole-cell currents recorded at −60 mV membrane potential from HEK293 cell co-expressing GluA1/A2$_{cryoEM}$ in response to 2-ms (green) or 1-s (blue) applications of 3 mM Glu alone or application of Glu in the continuous presence of 30 μM CTZ (orange). The inset shows closeup superposition of currents in response to 2-ms and 1-s applications of Glu alone. **b** Superposition of GluA1/A2$_{cryoEM}$-mediated currents recorded using a two-pulse protocol, in which the initial 0.1-s application of Glu was made to produce steady-state desensitization and the second application repeated after allowing the channels to recover from desensitization for different time. The envelope of the peak currents evoked by the series of second applications gives the time course of recovery from desensitization. The red curve illustrates the Hodgkin-Huxley equation fit of the peak current amplitude, with the time constant of recovery from desensitization,

$\tau_{RecDes} = 75 \pm 4$ ms, and $m = 1.02 \pm 0.06$. **c–f** Time constants of desensitization (**c**, $\tau_{Des}$), deactivation (**d**, $\tau_{Deact}$) and recovery from desensitization (**e**, $\tau_{RecDes}$), and the fraction of non-desensitized channels (**f**, $I_{SS}/I_{Max}$) measured for currents recorded from HEK293 cells transfected with wild-type GluA1/A2 (blue) and GluA1/A2$_{cryoEM}$ (magenta). The asterisk indicates a statistically significant difference (two-sided two-sample $t$-Test, the significance is assumed if $P < 0.05$). Data are mean ± SEM, source data are provided. For $\tau_{Des}$ (**c**), the number of independent experiments, $n = 11$ for GluA1/A2$_{WT}$ and n = 17 for GluA1/A2$_{cryoEM}$. The probability for the two-sided two-sample $t$-Test, $P = 0.00032$. For $\tau_{Deact}$ (**d**), $n = 7$ for GluA1/A2$_{WT}$, $n = 10$ for GluA1/A2$_{cryoEM}$, and $P = 0.85$. For $\tau_{RecDes}$ (**e**), $n = 8$ for GluA1/A2$_{WT}$, $n = 8$ for GluA1/A2$_{cryoEM}$, and $P = 0.95$. For $I_{SS}/I_{Max}$ (**f**), $n = 9$ for GluA1/A2$_{WT}$, $n = 12$ for GluA1/A2$_{cryoEM}$, and $P = 0.13$.

Figs. 1–3), with clearly resolved densities for all three transmembrane helices (M1, M3, and M4) and the re-entrant M2 loop (Supplementary Fig. 4a, b), ZK, Glu, (R,R)−2b, and Quis (Supplementary Fig. 4c–f). The overall architecture of all three GluA1/A2 core conformations revealed the canonical three-layer, Y-shaped domain organization characteristic of iGluRs[27,32,34,35], comprising the amino-terminal domain (ATD), which mediates receptor assembly and trafficking[45,46]; the ligand-binding domain (LBD), which binds agonists and competitive antagonists[47,48]; and the transmembrane domain (TMD)[29,49,50], which houses the ion channel pore. The tetramer adopts a dimer-of-dimers organization that defines four non-equivalent subunit positions, named consecutively A, B, C, and D[32], with subunits GluA1 in the A/C positions and GluA2 in the B/D positions, unambiguously defined based on the appearance of density for carbohydrates attached to glycosylation sites unique for GluA1 and GluA2 (Supplementary Fig. 4g–j). In the ATD layer, subunits A/B and C/D form local dimers, with subunits B and D

being proximal while A and C distal. In contrast, in the LBD layer, the pairing is rearranged to A/D and B/C, with subunits A and C being proximal and B and D distal, producing a domain-swapped architecture between extracellular layers.

**Closed-state structure of heteromeric GluA1/A2 AMPA receptor**

To resolve the structure of the GluA1/GluA2 AMPAR core in the closed resting state, we subjected our purified protein to cryo-EM in the presence of the competitive antagonist ZK (Fig. 2a,d). The GluA1/A2$_{ZK}$ structure revealed hallmark characteristics of non-conducting, antagonist-bound AMPAR structures[28,32,34,35]. Due to the flexibility of the ATD layer that worsens the overall resolution of the full-length receptor map and given its limited contribution to AMPAR function[51–53], we focused our structural analysis on the LBD-TMD region. The LBD layer assembles as a dimer-of-dimers, where A/D and B/C dimer pairs are oriented symmetrically around the axis of overall

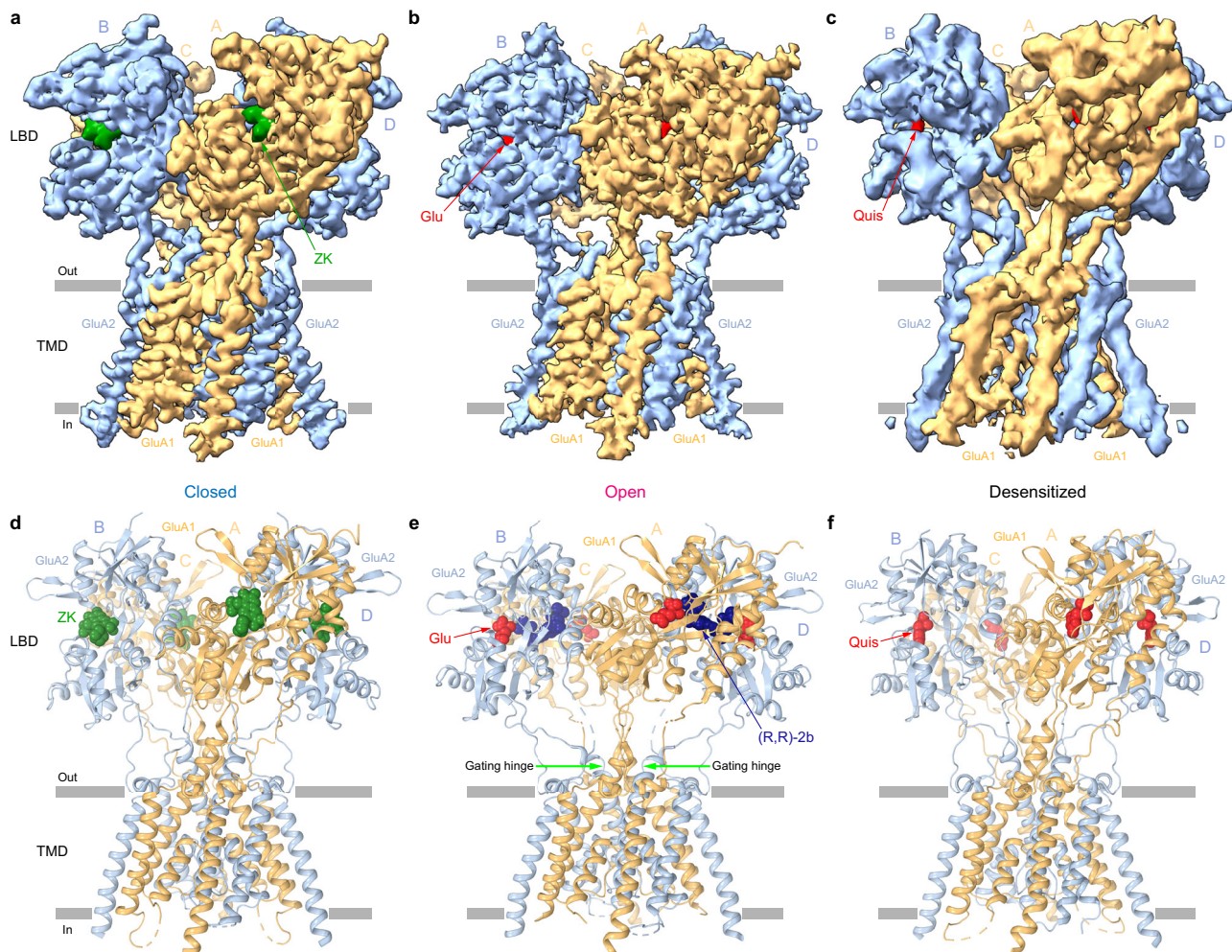

**Fig. 2 | Cryo-EM structures of heteromeric GluA1/A2 receptors.** 3D reconstructions (**a**–**c**) and structural models (**d**–**f**) of GluA1/A2$_{cryoEM}$ in the antagonist ZK-bound closed (**a**, **d**), agonist Glu and PAM (R,R)−2b-bound open (**b**, **e**), and agonist Quis-bound desensitized (**c**, **f**) states, viewed parallel to the membrane, with density and models for GluA1 (A/C) and GluA2 (B/D) subunits colored light orange and blue, respectively, ZK green, Glu and Quis red, and (R,R)−2b dark blue. The gating hinges in the structural model of the open state are indicated by green arrows.

two-fold rotational symmetry. The LBD adopts a bilobed "clamshell" architecture, consisting of an upper lobe (D1) and lower lobe (D2). In this structure, each LBD clamshell is widely open, with a single molecule of ZK bound in the cleft between the D1 and D2 lobes. Each LBD dimer has a tight D1-D1 interface displaying 20 Å of distance between the Cα atoms of A737 in GluA1 and S741 in GluA2, with a compact interface between the lower D2 lobes measured at 18 Å between Cα atoms of S631 (GluA1) and S635 (GluA2) (Fig. 3a). This short distance between the D2 lobes within individual LBD dimers ensures that the TMD-facing surface of the LBD tetramer remains tight (Fig. 3b), with a 24-Å distance between the GluA1 subunits (S631 residues in subunits A and C) and 55-Å distance between the GluA2 subunits (S635 residues in subunits B and D) (Fig. 3c) and leaves the LBD-TMD linkers in a relaxed conformation. This compact arrangement of subunits continues into the TMD, where the upper portion of all M3 helices, which form the channel gate, display non-kinked conformations at the gating hinge, located at A614 in GluA1 and A618 in GluA2 (Fig. 3c). The TMD, viewed from the extracellular side, shows a narrow pore organization (Fig. 3d), which suggests a non-conducting conformation of the ion channel. Additionally, the elusive M2 helices and the selectivity filter peptides are well resolved, completing the pore-lining segments of the TMD. This arrangement is typical for what has been previously reported in closed-state structures of GluA2 homomer alone[27,29,31–35,50] and other AMPARs in complex with auxiliary subunits[26,41,54–60].

## Open-state structure of GluA1/A2 and mechanism of activation

To investigate activation of the heteromeric GluA1/A2 core, we solved the structure of GluA1/A2 in the presence of Glu and PAM (R,R)−2b, which blocks receptor desensitization and increases steady-state current amplitude[61]. The resulting GluA1/A2$_{Glu+RR2b}$ structure displayed features similar to those observed in previously reported open-state AMPAR structures[29,58–60,62–64]. LBD clamshells are closed around the agonist, with one molecule of Glu bound between the D1 and D2 lobes of each individual LBD clamshell, and one molecule of (R,R)-2b bound at the LBD dimer interface. Strengthened by (R,R)-2b binding, the D1-D1 dimer interface remains intact with the separation of D1 lobes maintained at approximately 20 Å. At the same time, the D2 lobe swings upward towards D1 in 26° and 25° rotations for GluA1 and GluA2, respectively, resulting in a 32 Å distance between D2 lobes in each dimer (Fig. 3e), an increase of 14 Å compared to the closed state (Fig. 3a). This agonist-induced separation of the D2 lobes of the LBD dimer represents the driving force of AMPAR activation, originally proposed in studies of homotetrameric AMPARs (Supplementary Fig. 5)[29,50,65].

A top-down view of the LBD layer in GluA1/A2$_{Glu+RR2b}$ reveals a dramatic expansion of the LBD tetramer, with the outward

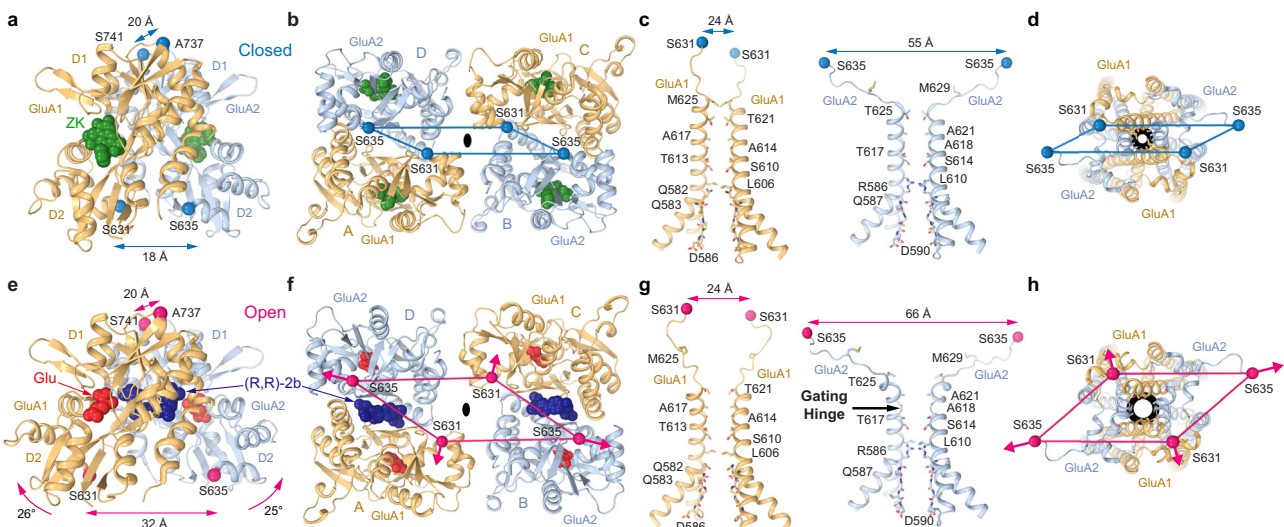

**Fig. 3 | Activation of heteromeric GluA1/A2 receptors.** Structures of LBD dimers (**a**, **e**), LBD tetramers (**b**, **f**), pore-forming elements in GluA1 (left) and GluA2 (right) subunits (**c**, **g**), and ion channels viewed extracellularly (**d**, **h**) in the closed (**a–d**) and open (**e–h**) states, with GluA1 (A/C) and GluA2 (B/D) subunits colored light orange and blue, respectively. Molecules of ZK, Glu, and (R,R)-2b are shown as green, red, and dark blue space-filling models. Closure of the individual LBD clamshells and expansion of the LBD layer during activation are indicated by one-sided pink arrows. Two-sided arrows indicate distances between the upper or lower LBD lobes within the LBD dimers or between the diagonal subunits. The gating hinge in the open state is indicated by black arrow. Black rings illustrate the extent of pore opening.

displacement of the gating M3-S2 linker attachments at S631 in GluA1 and S635 in GluA2 (Fig. 3f), resulting in 10-Å lengthening of the S635-S635 (B/D) distance to 66 Å from 55 Å in GluA1/A2$_{ZK}$, while maintaining a distance of 24 Å in S631-S631 (A/C) (Fig. 3c, g). Thus, the M3 helix in GluA1 unwinds by one helical turn, while the M3 helix in GluA2 kinks at the gating hinge A618 (Fig. 3c, g). Consequently, a dilated channel pore is observed along the entirety of the channel gate, consistent with an open, conducting conformation (Fig. 3h), similar to previously reported open-state structures of AMPARs[29,50,56,58–60,62].

## Activation of GluA1/A2 in the presence of auxiliary subunits

During 3D classification of purified GluA1/A2 in the presence of Glu and (R,R)-2b, we identified a subpopulation of particles containing four extra membrane-spanning densities peripherally surrounding the TMD (Supplementary Fig. 2). Local refinement and symmetry expansion of the LBD-TMD region of these particles produced a ~3.5 Å resolution map, which had high resemblance with maps representing complexes of AMPAR with members of CNIH family of auxiliary subunits[30,55,56] (Fig. 4a). Since we did not transduce our HEK 293 cells with CNIH viruses, we performed mass spectrometry (MS) analysis of the purified protein sample to identify which specific paralogs of endogenous human CNIHs, CNIH1-4, can be represented by the cryo-EM density. We found that all four paralogs were present in our sample and apparently carried out from HEK 293 cells throughout the purification, with CNIH4 being most abundant, followed by CNIH1, CNIH3 and then finally CNIH2 (Supplementary Data 1). Because of the appearance of density for N-terminal residues that are present in CNIH1-3 but not in CNIH4, and a few map locations that demonstrate slight favor for the shape of CNIH1 residues (Supplementary Fig. 6), we modelled the density with CNIH1 (Fig. 4b). However, given that the MS analysis revealed the presence of all four CNIHs in our protein sample and ambiguity of cryo-EM map interpretation, we refer to our structure as GluA1/A2-CNIH$_{Glu+RR2b}$, implying that all four CNIHs, CNIH1-4, can contribute to the resolved cryo-EM density.

CNIHs are known as auxiliary subunits that form complexes with AMPARs in the brain[38]. CNIH2 and CNIH3 enhance AMPAR trafficking, slow deactivation and desensitization kinetics, and increase the channel open probability[38,66,67]. CNIH2 has been structurally characterized with heteromeric GluA1/A2 AMPAR cores bound to transmembrane AMPAR regulatory protein (TARP) γ8[56]. This structure revealed a 2:1 stoichiometry of CNIH2 to the receptor core, where two CNIH2 subunits occupy the positions between M1 of GluA1 and M4 of GluA2 of heteromeric receptors and two TARP subunits occupy the positions between M1 of GluA2 and M4 of GluA1. However, we observe a 4:1 stoichiometry of GluA1/A2 core bound to four CNIH protomers surrounding the receptor core, alike in the previously published structure of homomeric GluA2 in complex with four CNIH3 subunits[30]. All CNIHs lack the bulky extracellular head that is a characteristic of TARPs, avoiding steric clashes with the LBDs, which likely allows them to bind to all four available sites around the GluA1/A2 TMD.

Similar to GluA1/A2$_{Glu+RR2b}$, GluA1/A2-CNIH$_{Glu+RR2b}$ contains one Glu molecule per closed LBD clamshell and one (R,R)−2b molecule per LBD dimer (A/D and B/C interfaces) (Fig. 4b). The presence of kinked M3 helices in subunits B and D and similar separation of the LBD dimer D2 lobes in both GluA1/A2$_{Glu+RR2b}$ (Fig. 3g) and GluA1/A2-CNIH$_{Glu+RR2b}$ suggest that these structures represent the open state of the core, independent of the presence or absence of CNIH. To assess structural changes that accompany activation, we examined the LBD architecture of the GluA1/A2 core alone with the same core bound to the γ8 subunit[62], both γ8 and CNIH2 subunits[56], and fully occupied by CNIHs. We measured the degree of clamshell closure for individual LBDs and quantified distances between the D2 lobes within LBD dimers (A/D and B/C) and across the LBD tetramer (B/D pair). All structures in the open state displayed increased clamshell closure (~25°) in A/C and B/D subunits relative to the closed state structure, GluA1/A2$_{ZK}$ (Fig. 4c), reflecting complete agonist occupancy. Similarly, all open state structures exhibited greater D2 lobe separation of 32 Å across LBD dimers (Fig. 4d) and across dimer pairs (65 Å distance between Cα's of S635 in GluA2 chains B and D) (Fig. 4e) compared to GluA1/A2$_{ZK}$ (18.5 Å and 55.4 Å, respectively). The similarity in LBD rearrangements among all open-state structures indicates that auxiliary subunits modulate AMPAR activation through mechanisms outside of the LBD.

## Pore architecture and water conductance in auxiliary subunit-bound GluA1/A2

Since auxiliary subunits differentially affect ion channel function but appear to have no effect on the LBD layer during activation (Fig. 4c–e), we questioned whether they induce structural changes within the TMD.

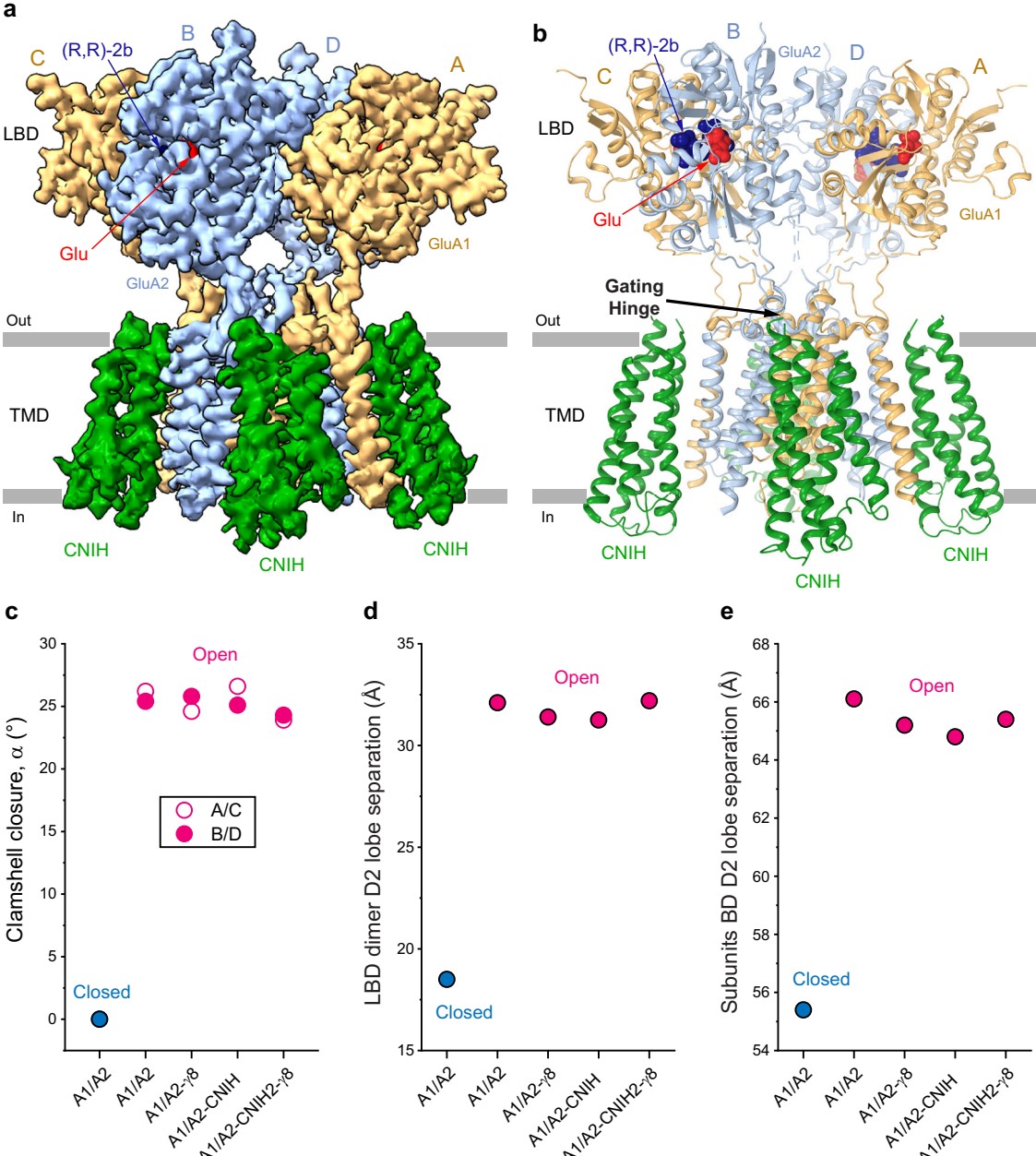

**Fig. 4 | Structural effects of auxiliary subunits on activation of heteromeric AMPA receptors. a–b** 3D reconstruction (**a**) and structural model (**b**) of GluA1/A2-CNIH$_{Glu+RR2b}$ in the Glu and (R,R)-2b bound open state, viewed parallel to the membrane, with density and models for GluA1 (A/C) and GluA2 (B/D) subunits colored light orange and blue, respectively, CNIH green, Glu red, and (R,R)-2b dark blue. Black arrow in **b** indicates kinking of the M3 helices in subunits B and D at the gating hinge. CNIH model is for human CNIH1. **c–e** Pink circles show parameters characterizing the open-state structures of GluA1/A2, GluA1/A2-γ8 (PDB ID: 7QHB), GluA1/A2-CNIH and GluA1/A2-CNIH2-γ8 (PDB ID: 7OCF), the angle of LBD clamshell closure (α, **c**), LBD dimer D2 lobe separation (distance between Cα's of S631 in GluA1 chain A and S635 in GluA2 chain D, **d**), and subunits B/D D2 lobe separation (distance between Cα's of S635 in GluA2 chains B and D, **e**). Parameters for the closed-state structure of GluA1/A2 (blue circles) are added as references.

Because we resolved the M2 re-entrant pore loop, we were able to measure the entire pore geometry in different activated-state structures and compared it to the pore geometry in the non-conducting state of the GluA1/A2 core (Fig. 5a–c). As expected, the narrowest pore was observed in the closed GluA1/A2$_{ZK}$ state structure (Fig. 5b, c). Interestingly, the pores of the active-state structures had different dimensions in the gate region, suggesting that auxiliary subunits may influence AMPAR water and ion conductance. To test this hypothesis, we compared the behavior of the closed-state GluA1/A2$_{ZK}$ structure with active-state structures of GluA1/A2, GluA1/A2-γ8 and GluA1/A2-CNIH2-γ8 in molecular dynamics (MD) simulations.

To characterize the pore state with regard to water conductance, we performed unrestrained simulation of the cryo-EM structure of GluA1/A2$_{ZK}$ and backbone-restrained simulations of GluA1/A2$_{Glu+RR2b}$, GluA1/A2-γ8$_{Open}$[62] and GluA1/A2-CNIH2-γ8$_{Open}$[56]. In these restrained simulations, all backbone heavy atoms (N, Cα, C, and O) of the heteromeric receptor core are kept harmonically restrained in place. This enabled the comparison of water conductance in conformations closest to their cryo-EM structures (see **Methods**). All open state structures showed evidence of water permeation (Fig. 5d), indicating full hydration of their pores. In contrast, GluA1/A2$_{ZK}$ did not display water permeation. GluA1/A2$_{Glu+RR2b}$ showed the lowest conductance rate,

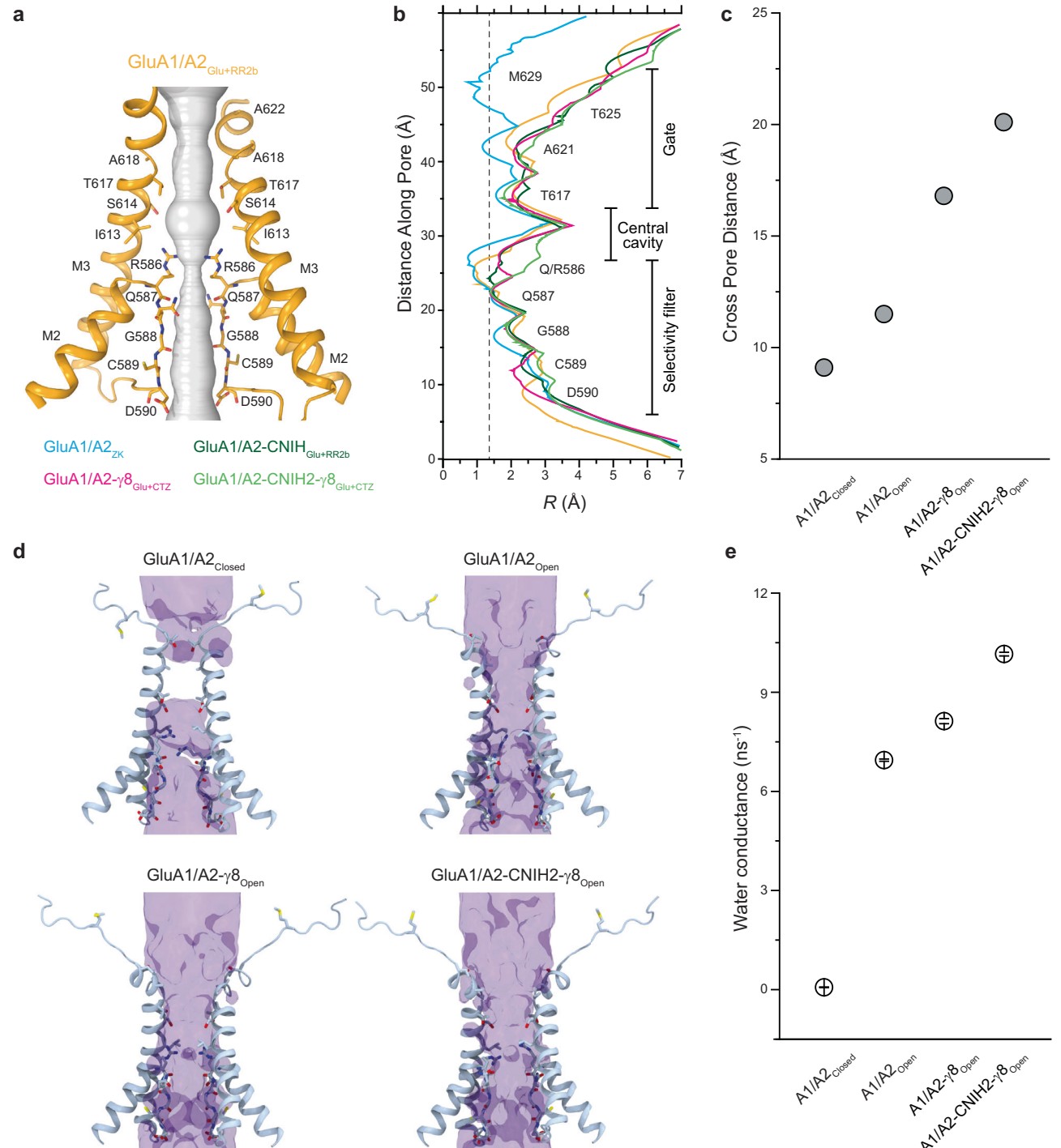

**Fig. 5 | Ion channel pore and water conductance. a** Pore-forming domains M2 and M3 in GluA1/A2$_{Glu+RR2b}$ with the residues lining pore shown as sticks. Only two (B and D) of four subunits are shown, with the front and back subunits (A and C) omitted for clarity. The pore profile is shown as a space-filling model (grey). **b** Pore radius for the closed-state structure GluA1/A2$_{ZK}$ (blue), and open-state structures GluA1/A2$_{Glu+RR2b}$ (orange), GluA1/A2-γ8$_{Glu+CTZ}$ (pink; PDB ID: 7QHB), GluA1/A2-CNIH$_{Glu+RR2b}$ (dark green), and GluA1/A2-CNIH2-γ8$_{Glu+CTZ}$ (light green; PDB ID: 7OCF) calculated using HOLE. The vertical dashed line denotes the radius of a water molecule, 1.4 Å. **c** Cross-pore distance between Cα atoms of A621 of GluA2 subunits (B and D) in GluA1/A2$_{ZK}$, GluA1/A2$_{Glu+RR2b}$, GluA1/A2-γ8$_{Glu+CTZ}$ (PDB ID: 7QHB), and GluA1/A2-CNIH2-γ8$_{Glu+CTZ}$ (PDB ID: 7OCF). **d** Water occupancy in MD simulations.

Channel pores with the integral occupancy by water molecules shown as violet continuum for MD simulations of GluA1/A2$_{ZK}$, GluA1/A2$_{Glu+RR2b}$, GluA1/A2-γ8$_{Glu+CTZ}$, and GluA1/A2-CNIH2-γ8$_{Glu+CTZ}$. Pore-forming segments M2 and M3 helices (GluA2 subunits B and D) are shown as blue ribbons, with residues lining the pore shown as sticks. Continuous water occupancy, an indication of water conductance, is observed in all structures, except closed-state structure GluA1/A2$_{ZK}$, which remains non-conducting throughout the simulations. **e** Average (mean) water conductance estimated from MD simulations of GluA1/A2$_{ZK}$, GluA1/A2$_{Glu+RR2b}$, GluA1/A2-γ8$_{Glu+CTZ}$, and GluA1/A2-CNIH2-γ8$_{Glu+CTZ}$. Error bars represent standard deviation of water permeation counts within each simulation.

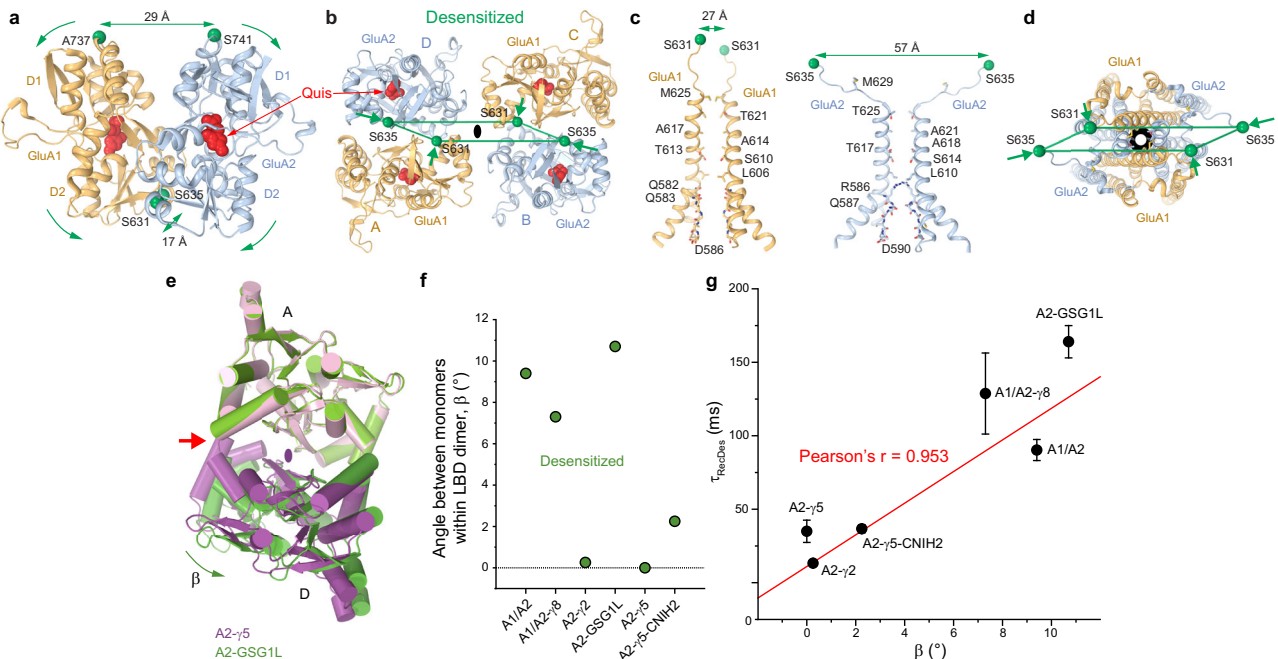

**Fig. 6 | Desensitization of heteromeric GluA1/A2 core and effects of auxiliary subunits.** Structures of LBD dimer (**a**), LBD tetramer (**b**), pore-forming elements in GluA1 (left) and GluA2 (right) subunits (**c**), and ion channel viewed extracellularly (**d**) in the desensitized state, with GluA1 (A/C) and GluA2 (B/D) subunits colored light orange and blue, respectively. Molecules of Quis are shown as red space-filling models. Relative rotation of the LBD clamshells within the LBD dimer accompanying D1-D1 interface rupture and constriction of the LBD layer during desensitization compared to the open state are indicated by one-sided green arrows. Two-sided green arrows indicate distances between the upper or lower LBD lobes within the LBD dimer or between the diagonal subunits. The black ring illustrates the extent of pore opening. **e** Superposition of LBD dimers from the desensitized-state structures of GluA2-γ5 (purple) and GluA2-GSG1L (green), viewed from the top, with the local axis of 2-fold rotational symmetry for GluA2-γ5 shown as a purple oval. The green arrow indicates rotation of subunit D LBD away from subunit A LBD in GluA2-GSG1L, accompanied by the loss of the A/D LBD dimer 2-fold rotational symmetry and the opening of a cleft between the protomers (red arrow). **f** The angle of deviation from the 2-fold rotational symmetry in the desensitized-state structures (β, green circles). **g** Correlation between β and the time constant of recovery from desensitization ($\tau_{RecDes}$). The values of $\tau_{RecDes}$ are mean ± SEM for GluA1/A2, GluA2-γ2, GluA2-GSG1L, GluA2-γ5 and GluA2-γ5-CNIH and mean ± SD for GluA1/A2-γ8, determined in the current study (GluA1/A2) or taken from previous studies[27,28,54,55,75] (also see Supplementary Fig. 9h legend).

whereas higher average water conductance rates were observed in the simulations of GluA1/A2-γ8_Open and GluA1/A2-CNIH2-γ8_Open, suggesting that CNIH2 and γ8 play a dominant role during activation of GluA1/A2 core (Fig. 5e and Supplementary Fig. 7a–d, g). Instant occupancies of water molecules at the gate residues T617, A621, and T625 were also higher for GluA1/A2-γ8_Open and GluA1/A2-CNIH2-γ8_Open, which is consistent with the average conductance rates (Supplementary Figs. 7e–g, 8). These results indicate that CNIH2 and γ8 increase the pore diameter, with wider pores associated with higher water conductance rates. Since the auxiliary subunits appear to have no effect on the LBD layer during activation (Fig. 4c-e), this also highlights the influence of auxiliary subunits on AMPAR transmembrane domain architecture and dynamics, which are critical for the receptor gating. Accordingly, small molecules perturbing AMPAR-auxiliary subunit TMD interfaces may represent promising drug candidates. In this regard, a prospective location for drug targeting is the extracellular part of the AMPAR-auxiliary subunit TMD interface, which is adjacent to the AMPAR extracellular collar involved in gating and hosts anti-epileptic drugs like perampanel[55,68,69]. For instance, such interface in the AMPAR-γ8 complex has already been targeted with several drug candidate molecules[70–72].

## Structural rearrangements of AMPARs during desensitization

To investigate conformational changes in the AMPAR core associated with desensitization, we solved the cryo-EM structure of GluA1/A2 in the presence of agonist Quis (GluA1/A2_Quis), which, similar to Glu, binds inside the LBD clamshell and causes its closure, yet has a higher affinity ($EC_{50} = 16\,\mu M$) for the receptor than the natural agonist Glu ($EC_{50} = 300\,\mu M$)[73]. GluA1/A2_Quis shows typical structural features that

characterize the desensitization mechanism of AMPAR proposed before (Fig. 6a–d and Supplementary Fig. 9a–f)[27,50,74]. Indeed, in contrast to receptor activation, during which the D1-D1 interface of LBD dimer remains intact, the LBD closure in GluA1/A2_Quis causes the D1-D1 interface rupture. As a result, the D1 lobes undergo separation with the distance of 29 Å between the Cα atoms of A737 in GluA1 and S741 in GluA2, while the D2 lobes remain close to each other with the distance of 17 Å between the Cα atoms of S631 in GluA1 and S635 in GluA2, like in the closed state (Fig. 6a). The lack of D2-D2 lobe separation, which drives pore dilation in the open state, ensures closed conformation of the ion channel, a characteristic of the desensitized state. Accordingly, the extracellular view of the LBD layer reveals no expansion of the LBD tetramer (Fig. 6b), which characterizes the open state (Fig. 3h). Accordingly, the distance between D2 lobes of the B/D diagonal elements, which is greatly increased in the open state to drive pore opening (65 Å, Fig. 3g), is now reduced to 57 Å, comfortably stabilizing the pore in the closed conformation. Indeed, the M3 helix is one helical turn longer in GluA1 (A/C subunits) and displays a non-kinked configuration at the gating hinge A618 in GluA2 (B/D subunits) (Fig. 6c), in contrast to that in the open-state structure GluA1/A2_Glu+RR2b (Fig. 3g). An extracellular view of the TMD highlights the constriction of the ion channel and the gate (Fig. 6d), and low water permeation rates observed in MD simulations further support a non-conducting pore characteristic of the desensitized state.

## Desensitization of AMPARs in the presence of auxiliary subunits

To assess how auxiliary subunits influence AMPAR desensitization, we compared key functional and structural characteristics of desensitization, focusing specifically on the kinetics of desensitization and

recovery from desensitization, as well as the ratio of steady-state to peak current amplitude for AMPARs in complex with TARPs γ2, γ5, and γ8, germline specific gene 1-like (GSG1L), and CNIH2, as these complexes have had their function extensively characterized[27,44,55,66,75,76] and structures in the desensitized state determined, including GluA1/A2-γ8 (PDB ID: 7QHH)[62], GluA2-γ2 (PDB ID: 9MRL)[58], GluA2-GSG1L (PDB ID: 7RZA)[54], GluA2-γ5 (PDB ID: 7RZ7)[54] and GluA2-γ5-CNIH2 (PDB ID: 8SSB)[39]. Receptors bound to γ2, γ8, and γ5 together with CNIH2 showed significantly increased values for the time constant of desensitization ($\tau_{Des}$) (Supplementary Fig. 9h), prolonging receptor activity and enhancing synaptic charge transfer, whereas GSG1L and γ5 (without CNIH2) had minimal effects on $\tau_{Des}$. Auxiliary subunits γ2, γ5, and γ5 with CNIH2 appear to accelerate the recovery from desensitization by reducing the corresponding time constant ($\tau_{RecDes}$), enabling receptors to rapidly return to a responsive state, whereas γ8 and GSG1L slow recovery from desensitization, prolonging the non-conducting state of the receptor (Supplementary Fig. 9g). TARPs γ2 and γ8 increase the steady state to peak current ratio ($I_{ss}/I_{max}$), which approximates the fraction of non-desensitized receptors in equilibrium, during prolonged Glu exposure (Supplementary Fig. 9i).

All desensitized-state structures displayed an increased separation of D1 lobes of the LBD dimers (Supplementary Fig. 9j) when compared to closed and open state structures, where D1 lobes are firmly in contact with one another (Fig. 3a, e). However, the distance between D2 lobes of B/D subunits in the desensitized state structures is more similar to the closed state (Supplementary Fig. 9k), emphasizing their non-conducting nature. Further analysis of LBD dimers revealed varying degrees of swinging (β) of subunit D (or B) clamshell away from subunit A (or C) clamshell within each dimer, resulting in a break of the local two-fold rotational symmetry. For instance, each dimer in the GluA2-γ5 structure (Fig. 6e, purple) maintains its two-fold rotational symmetry, whereas in GluA2-GSG1L (Fig. 6e, green), one LBD monomer swings away from another by $\beta = 10.7°$, indicating a complete loss of two-fold symmetry by the LBD dimer. Different desensitized state structures displayed varying β values, except for GluA2-γ2 and GluA2-γ5, which have preserved ~two-fold rotational symmetry of each LBD dimer (Fig. 6f). Strikingly, we observed a strong positive correlation (Pearson's $r = 0.953$) between β and $\tau_{RecDes}$, indicating that stronger distortion of the LBD dimer two-fold rotational symmetry is associated with larger $\tau_{RecDes}$, or slower recovery from desensitization (Fig. 6g). Complexes with minimal LBD dimer symmetry distortion (e.g., GluA2-γ2 and GluA2-γ5) exhibited the fastest recovery, whereas those with pronounced relative rotation of monomers within the LBD dimer and loss of the LBD dimer two-fold rotational symmetry (e.g., GluA2-GSG1L, GluA1/A2-γ8, and GluA1/A2) recovered most slowly. This suggests that stronger deviation of LBD dimer from two-fold rotational symmetry increases the energetic barrier for returning to the resting state, slowing the rate of receptor reactivation, which has been suggested previously[27].

## Discussion

AMPARs, the primary mediators of fast excitatory neurotransmission in the CNS, are essential for higher-order brain functions such as motor coordination, learning, and memory. Among AMPAR assemblies, the heterotetrameric GluA1/A2 receptor is the predominant form in the forebrain and hippocampus[25,77,78]. In this study, we present structural and functional analysis for the GluA1/A2 core in the closed, open, and desensitized states, without auxiliary subunits, as well as GluA1/A2-CNIH in the open state. The GluA1/A2 core adopts the canonical "Y-shaped" architecture, with GluA1 occupying the A/C and GluA2 the B/D positions in a dimer-of-dimers configuration. All three gating states resolve the full TMD, including the previously elusive M2 re-entrant loop that lines the selectivity filter (Fig. 2a−c)[32,34,35]. It has been suggested that the M2 re-entrant loop is stabilized by the presence of auxiliary subunits surrounding the TMD[26,28]. Consistent with this idea,

previous studies of homomeric GluA2, without auxiliary subunits, have lacked the M2 re-entrant pore loop[32,34,35,79], however the entire TMD is resolved in our study. One possible explanation is that these studies used the Q/R unedited Q586R construct, whereby Q586 lines the selectivity filter tip at all 4 positions (A-D). In contrast, heteromeric GluA1/A2 receptors include Gln at the Q/R site in GluA1 (Q582 at positions A and C) and Arg in GluA2 (R586 at positions B and D). This alternating configuration of Q and R at the tip of the re-entrant loop may be key to stabilizing the loop in the absence of auxiliary subunits.

Structural transitions from closed to open states, driven by Glu-induced clamshell closure in the LBD, propagate through the M3-S2 linkers to the channel gate, inducing pore opening. Structural comparisons across heteromeric GluA1/A2 AMPAR open state assemblies demonstrate that auxiliary subunits do not significantly alter LBD conformation. Instead, their primary impact is on the TMD, where CNIH2 and γ8 increase the pore radius and enhance water permeability, shown in our MD simulations (Fig. 5c−e). These findings suggest that auxiliary subunits fine-tune activation of heteromeric AMPARs by stabilizing the open state conformation without affecting ligand-driven LBD changes.

We further show the extent to which auxiliary subunits influence the dynamics of desensitization. The GluA1/A2 core possesses local asymmetry in the LBD layer during desensitization, measured by the rotation angle (β) between LBD dimer protomers. Remarkably, we find that β correlates with slower recovery from desensitization ($\tau_{RecDes}$) (Fig. 6g). Greater asymmetry suggests an increasing energetic barrier for transitioning back to the activated state. These trends align with prior studies on homomeric GluA2 receptors, in which greater LBD distortion was linked to prolonged desensitization[27,54]. Comparable results were observed for KARs, where the distortion of LBD dimers is even more pronounced, accompanied by a slower recovery from desensitization[80–83]. Together, these results support a model in which LBD dimer symmetry serves as a key determinant of gating kinetics, with auxiliary proteins precisely regulating transitions between functional states.

The ATD was only resolved in the open (GluA1/A2$_{Glu+RR2b}$) and desensitized (GluA1/A2$_{Quis}$) states (Supplementary Fig. 3), likely due to conformational flexibility, as 2D class averages suggest pronounced mobility in the closed state. Interestingly, the resolved ATD in the open and desensitized structures closely resembles that of GluA2 homomers[27,29,32,34], whereas the dynamic ATD behavior in closed conditions aligns more with the GluA1 homomer[59] and GluA1/A4 heteromers[84]. These results suggest that Ca2+-permeable subunits (GluA1, A3, and A4) increase ATD flexibility in a state-dependent manner. In addition, although all GluA1/A2 structures were derived from the same experimental preparation, CNIH was only resolved in the open state, and not in the resting or desensitized states. This contrasts with previous reports where CNIH2 remained bound to homomeric GluA2-γ5 in the resting and desensitized states[55] or heteromeric GluA1/A2-γ8[56] in the resting and open states, indicating that CNIH binding may be stabilized by a combination of subunit-, state-, or auxiliary subunit co-assembly-dependent manner. Further structural investigation into CNIH-bound AMPAR assemblies is required to address these questions.

A key limitation of our study is the incomplete structural characterization of auxiliary subunits within heteromeric complexes[26,41,56,57,62,84]. While high-resolution structures of GluA1/A2 are now available in complex with CNIH2 and/or γ8[41,56,62], which are classified as potentiating auxiliary subunits, structures of AMPARs in complex with other classes of auxiliary subunits, such as inhibitory type II TARPs (γ5 or γ7), GSG1L, CKAMPs, or SynDIGs, are missing. Moreover, most existing structural data derive from homomeric GluA2 assemblies[27–30,50,54,55,58,59,63,64,85]. To fully understand the diversity of AMPAR modulation, future studies must resolve high-resolution structures of other heteromeric combinations (e.g., GluA2/A3 and

GluA1/A2/A3), in the absence and presence of a broader range of auxiliary partners.

In summary, our work provides a structural and mechanistic framework for understanding how auxiliary subunits reshape both the asymmetry and gating transitions of the heterotetrameric GluA1/A2 AMPAR core. By resolving the full TMD, including the M2 loop, and identifying subtype-specific structural flexibility in the ATD, we highlight how subunit identity and auxiliary partners differentially sculpt AMPAR function. These findings offer molecular insight into the structural underpinnings of synaptic plasticity and establish a foundation for future efforts to decode the complete diversity of AMPAR assemblies and their roles in neurological health and disease.

## Methods

### Constructs for large-scale protein expression
Heteromeric GluA1/A2 was prepared using rat GluA1 flip isoform (rGRIA1, UniProt P19490-2), with 6 residues removed within the ATD-LBD linker (A374 – A379) and 66 residues removed from the C-terminus (F824 – L889), and rat GluA2 flip isoform, as described in previous studies[32] (rGRIA2, UniProt P19491-2), but instead with an edited R at the Q/R-site (R586). GluA1 and GluA2 were both introduced into pEG BacMam vector for baculovirus-based protein expression in mammalian cells[86]. These vectors include a thrombin cleavage site (LVPRG) at the C-terminus, mCherry and octa-Histidine affinity tag (8X His) or eGFP and Strep-II affinity tag (WSHPQFEK), for GluA1 and GluA2, respectively.

### Protein expression and purification
The GluA1/A2 expression and purification process followed previously established procedures with slight modifications[55]. Briefly, P1 and P2 viruses produced in Sf9 cells were added to HEK 293S GnTI⁻ cells at a ratio of 1:3 (A1:A2), which were incubated at 37 °C with 7% $CO_2$. 12–15 h post-transduction, 10 mM sodium butyrate was added to the cells, and the temperature was then lowered to 30 °C with 5% $CO_2$. The cultured cells were harvested 40–60 h post-transduction through low-speed centrifugation (5,500 g, 10 min), followed by a wash with 1X PBS pH 8.0 and another centrifugation (5500 × g, 15 min). Cells were subsequently lysed in the buffer containing 150 mM NaCl, 20 mM Tris pH 8.0, and protease inhibitors (0.8 μM aprotinin, 2 μg/ml leupeptin, 2 μM pepstatin A, and 1 mM phenylmethylsulfonyl fluoride, PMSF) using a Misonix Sonicator with 15 s "on" at the amplitude 8 followed by 15 s "off" for three min; this program was repeated three times for optimal cell lysis, under constant stirring on ice. The lysate was centrifuged for 15 min at 5500 × g to remove unbroken cells and cell debris, and the supernatant was subjected to ultracentrifugation (186,000 × g, 40 min) to pellet the cell membranes. The membrane pellet was mechanically homogenized and solubilized for 3 h at 4 °C in the buffer containing 150 mM NaCl, 20 mM Tris-HCl pH 8.0, 1% digitonin (Cayman Chemical Company, 14952), and the same protease inhibitors as were used in the lysis buffer. The insoluble material was removed by another round of ultracentrifugation (186,000 × g, 40 min). The supernatant was combined with pre-equilibrated streptavidin-linked resin (2 ml) and rotated for 1–2 h at 4 °C. Subsequently, the protein-bound resin was washed with 15 ml buffer containing 150 mM NaCl, 20 mM Tris-HCl pH 8.0, and 0.05% digitonin. The protein was eluted with 15 ml of the same buffer supplemented with 2.5 mM D-desthiobiotin. The eluted protein was then combined with pre-equilibrated Ni-NTA resin (2 ml) and rotated for 12–15 h at 4 °C. The protein-bound resin was washed with 15 ml buffer containing 150 mM NaCl, 20 mM Tris-HCl pH 8.0 and 0.05% digitonin supplemented with 25 mM imidazole. The protein was eluted by 15 ml of the same buffer supplemented with 250 mM imidazole. The eluted protein was concentrated and subjected to thrombin digestion (1:200 w/w) at 22 °C for 90 min to remove mCherry, eGFP, octa-His, and Strep-II affinity tags. The digest reaction was then injected into a Superose 6 Increase 10/

300 GL size-exclusion chromatography column (Cytiva) pre-equilibrated with a buffer containing 150 mM NaCl, 20 mM Tris-HCl pH 8.0, and 0.05% digitonin. The peak fractions corresponding to tetrameric GluA1/A2 were pooled, concentrated to 5–6 mg/ml (~60 μM), and used for cryo-EM sample preparation. All procedures, unless otherwise specified, were carried out at 4 °C.

### Cryo-EM sample preparation and data collection
UltrAuFoil R 0.6/1.0, 300 mesh gold grids (EMS, Morrisville, NC) were used for sample preparation. Before sample application, the grids were treated in a PELCO easyGlow cleaning system (Ted Pella, 25 s, 15 mA) to make their surface hydrophilic. Purified protein was supplemented with 100 μM ZK for GluA1/A2$_{ZK}$, 500 μM (R,R)-2b for GluA1/A2$_{Glu+RR2b}$, or 5 mM Quis for GluA1/A2$_{Quis}$, and incubated for 20 min on ice before plunge freezing. Samples for GluA1/A2$_{Glu+RR2b}$ were additionally spiked with 5 mM Glu immediately before plunge freezing. Grids were made using a Vitrobot Mark IV (Thermo Fisher Scientific) set at 100% humidity and 4 °C, blot time of 2 s, blot force of 3, and a wait time of 30 s. The grids were imaged on a Titan Krios transmission electron microscope (Thermo Fisher Scientific) operating at 300 kV equipped with a post-column GIF Quantum energy filter with slit width set to 20 eV and Gatan K3 (Gatan) direct electron detection camera using Leginon 3.5 software. We collected 14,743 movies (0.826 Å/pixel) for GluA1/A2$_{ZK}$, 20,912 movies (0.825 Å/pixel) for GluA1/A2$_{Glu+RR2b}$, and 13,380 movies (1.35 Å/pixel) for GluA1/A2$_{Quis}$ across a defocus range of −0.8 to −2.0 μm.

### Image processing
Data was processed using cryoSPARC 4.4.1 software[87]. For GluA1/A2$_{ZK}$, movie frames were aligned using patch motion correction algorithm. Contrast transfer function (CTF) estimation was performed using patch CTF estimation. Approximately 1000 micrographs were used for blob picking and extraction, followed by multiple rounds of 2D classification to create templates for template-based picking and training a Topaz model[88]. After combining particles from template-based and Topaz pickers, duplicates were removed, yielding a stack of 5,527,903 particles, which were extracted with a 448-pixel box size, down sampled to a 128-pixel box size. Reference-free 2D classification yielded a stack of 975,915 particles, which were subjected to heterogeneous refinements with one reference class created using ab-initio reconstruction and three automatically generated "garbage" classes, the best set of 565,171 particles were subjected to non-uniform refinement, yielding a 3.69 Å resolution map. As movies were obtained by beam tilt to optimize high-throughput data collection, beam tilt groups were applied to the CTF-estimated micrographs, which were then applied to the particles. Next, global CTF and non-uniform refinement were run again to obtain consensus refinement at 3.54 Å resolution. These particles were then subjected to reference-based motion correction, and further 3D classification without alignment. The final map (219,104 particles) of LBD-TMD was refined using C2 symmetry to 3.43 Å resolution, with local refinements of symmetry expanded (C2) particles of the LBD and TMD using C1 symmetry, both at 3.34 Å resolution. In order to aid model building of the complex, composite maps were generated by aligning both local maps to the global reconstruction, for both positions within the C2 symmetric complex, and then combining them by taking the maximum value at each voxel using UCSF ChimeraX (v.1.9)[89].

For GluA1/A2 in the presence of Glu and (R,R)−2b, all processing steps were similar to those used for GluA1/A2$_{ZK}$. Briefly, 5,802,290 particles picked from template-based and Topaz picker were extracted with a 440-pixel box size and down sampled to a 128-pixel box size. After several rounds of 2D and 3D cleanup, the best 704,982 particles were subjected to non-uniform refinement, yielding a 3.60 Å reconstruction. Beam tilt groups were applied to the CTF-estimated micrographs, which were then applied to the particles from non-uniform

refinement. Next, global CTF and non-uniform refinement were run to obtain a consensus refinement at 3.27 Å resolution. These particles were then subjected to reference-based motion correction, and the resulting pool of particles underwent 3D classification without alignment into 3 classes, which unexpectedly yielded one class of GluA1/A2 with co-purified CNIH (93,743 particles). Particles representing GluA1/A2 without CNIH were subjected to the non-uniform refinement with C2 symmetry applied and classified into 3 classes using a reference-free 3D classification algorithm. The total of 131,148 particles from the best two classes underwent refinement using non-uniform refinement with C2 symmetry imposed, resulting in a 3.20 Å reconstruction (GluA1/A2$_{Glu+RR2b}$ of the LBD-TMD. Local refinement of the LBD and TMD domains yielded maps at 3.13 and 3.81 Å resolution, respectively. Similarly, particles representing GluA1/A2 with CNIH yielded a consensus refinement of 3.75 Å resolution of the LBD-TMD (GluA1/A2-CNIH$_{Glu+RR2b}$). Local refinements of LBD and TMD yielded maps at 3.51 and 3.55 Å resolution, respectively.

For GluA1/A2$_{Quis}$, all processing steps were similar to those used for GluA1/A2$_{ZK}$. The total of 1,154,987 particles picked from template-based and Topaz picker were extracted and down sampled to a 128-pixel box size. After several rounds of 2D classification, 313,554 particles were re-extracted unbinned and subjected to heterogeneous refinement with one reference class created using ab-initio reconstruction and three automatically generated "garbage" classes. The best set of 251,523 particles was subjected to non-uniform refinement, yielding a consensus map at 4.91 Å resolution. Local refinement of the ATD (87,480 particles), LBD (89,333 particles), and TMD (32,669 particles) separately yielded maps at 4.28 Å, 4.59 Å, and 4.57 Å resolution, respectively.

Finally, the combine_focused_maps algorithm implemented in Phenix was used to create composite maps from LBD and TMD local refinements for GluA1/A2$_{ZK}$ and GluA1/A2-CNIH$_{Glu+RR2b}$ (Supplementary Figs. 1–2), and from ATD, LBD, and TMD local refinements for GluA1/A2$_{Glu+RR2b}$ and GluA1/A2$_{Quis}$ (Supplementary Figs. 2–3). The reported resolution for the final maps was estimated in cryoSPARC using the gold standard Fourier shell correlation (GSFSC) of FSC = 0.143 criterion. Cryo-EM densities were visualized using UCSF ChimeraX (v.1.9)[89]. Structural biology applications employed in this project adhered to and were configured by SBGrid.

## Model building and refinements
For all structures, molecular models of GluA1 and GluA2 were built in Coot (0.9.8.1)[90] using cryo-EM density as well as structures 3SAJ and 5WEO as guides. Structures of CNIH1-4 were built in Coot using AlphFold models as guides. All models were real space refined in Phenix (1.18)[91] and visualized in ChimeraX (1.9)[89] or Pymol 2.5.2 (The PyMOL Molecular Graphics System, version 2.0, Schrödinger). Domain rotations were determined using the DynDom server (http://dyndom.cmp.uea.ac.uk/dyndom/). The pore radius was calculated using HOLE (2.1)[92].

## Patch-clamp recordings
DNA encoding full-length and engineered versions of GluA1 and GluA2 were introduced into a pIRES plasmid for expression in eukaryotic cells that were engineered to produce green fluorescent protein via a downstream internal ribosome entry site[34]. HEK 293 cells (ATCC, CRL-1573) grown on glass coverslips in 35 mm dishes were transiently transfected with 1–5 µg of the plasmid DNA using Lipofectamine 2000 Reagent (Invitrogen). Recordings were made 24 to 72 h after transfection at room temperature. Currents from whole cells, typically held at a −60 mV potential, were recorded using Axopatch 200B amplifier (Molecular Devices, LLC), filtered at 5 kHz, and digitized at 10 kHz using low-noise data acquisition system Digidata 1440 A and pCLAMP 10.2 software (Molecular Devices, LLC). The external solution contained (in mM): 140 NaCl, 2.4 KCl, 4 CaCl$_2$, 4 MgCl$_2$, 10 HEPES pH 7.3

and 10 glucose; 7 mM NaCl was added to the extracellular activating solution containing 3 mM Glu to improve visualization of the border between two solutions coming out of a two-barrel theta glass pipette, which allowed its more precise positional adjustment for faster solution exchange. The internal solution contained (in mM): 150 CsF, 10 NaCl, 10 EGTA, 20 HEPES pH 7.3. Rapid solution exchange was achieved by mounting the two-barrel theta glass pipette on a piezo-electric translator. Typical 10-90% rise times were 200-300 µs, as measured from junction potentials at the open tip of the patch pipette after recordings. Data analysis was performed using Origin 2023 software (OriginLab Corp.). Recovery from desensitization recorded using the two-pulse protocol was fitted with the Hodgkin-Huxley equation[93]: $I = (I_{max}^{1/m} - (I_{max}^{1/m} - 1) \times \exp(-t/\tau_{RecDes}))^m$, where $I$ is the peak current at a given interpulse interval, $t$, $I_{max}$ is the peak current at long interpulse intervals, $\tau_{RecDes}$ is the recovery time constant and $m$ is an index that corresponds to the number of kinetically equivalent rate-determining transitions that contribute to the recovery time course.

## System preparation for MD simulations
The open state cryo-EM structures of GluA1/A2$_{Glu+RR2b}$, GluA1/A2-γ8$_{Open}$ (PDB ID: 7QHB)[62], and GluA1/A2-CNIH2-γ8$_{Open}$ (PDB ID: 7OCF)[56] and the closed state structure of GluA1/A2$_{ZK}$ were used as initial atomic coordinates for MD simulations. All cryo-EM structures used in the MD simulations were composed of the LBD and TMD. Missing residues in the LBD-TMD linkers and M1-M2 intracellular loops were modeled using MODELLER[94]. A total of 200 models were generated, and the ones with the lowest Discrete Optimized Protein Energy (DOPE) scores were selected. The modeled loops were further refined before incorporation into the simulation systems. For all MD systems, we included the proteins, ligands in the LBDs, and the modulators in the LBD dimer interfaces from the cryo-EM structures and removed all other molecules. CHARMM-GUI membrane builder[95,96] was used to construct each simulation box by inserting the protein into a POPC bilayer and solvating it with TIP3P water molecules and 150 mM KCl. The systems were set up for molecular dynamics simulations using the "tleap" module of the AmberTools20 package[97]. All the ligands were parametrized using the general AMBER force field (GAFF2)[98].

## MD simulation protocols
The Pmemd.cuda program of the Amber20 MD software package was used for all MD simulations[99]. Amber FF99SB–ILDN[100] force field parameters were used for protein and ions, TIP3P model for water, and Lipid14 force field[101] parameters for lipids. Hydrogen mass repartitioning (HMR) was applied to all systems using parmed[102] to redistribute mass from heavy atoms to bonded hydrogen atoms, increasing hydrogen masses to ~3.024 amu. This allowed the use of a 4 fs integration timestep while maintaining stability.

All equilibration and production simulations were performed in NPT ensemble at 300 K temperature and 1 bar pressure with anisotropic pressure coupling. The temperature was controlled using Langevin thermostat with a collision frequency of 1 ps$^{-1}$ and the pressure was controlled with a Berendsen barostat with a 1 ps relaxation time, as implemented in Amber20. All covalent bonds involving hydrogen atoms were constrained using the SHAKE algorithm[103], with the integration time step of 4 fs. The long-range electrostatic interactions were approximated using the particle mesh Ewald (PME) method[104], with a non-bonded interaction cutoff radius of 10 Å. Periodic boundary conditions were applied in all three dimensions.

We employed two different simulation protocols. The first protocol (equilibration and production simulations) was performed as follows. Each system was initially energy-minimized with harmonic restraints applied on all protein Cα and ligand heavy atoms. This was followed by the equilibration of water and ions at constant volume MD simulations as the temperature was gradually increased from 0 to 300 K, with all protein, ligand and lipid heavy atoms harmonically

restrained at their energy minimized positions with the force constant of 40 kcal mol$^{-1}$ Å$^{-2}$. The systems were then equilibrated at constant pressure for 100 ns, gradually releasing the restraints on the protein and ligands to 0.5 kcal mol$^{-1}$ Å$^{-2}$. Production simulations were then conducted for approximately 100 ns with all restraints removed. This protocol was used for the simulation of GluA1/A2$_{ZK}$.

The second protocol (long equilibration simulations) was performed as follows. To enable the comparison of water conductance between structures in conformations closely resembling their respective cryo-EM structures, we performed ~500 ns equilibration simulations where all the backbone heavy atoms of the heteromeric receptor core are kept harmonically restrained by a force constant of 10 kcal mol$^{-1}$ Å$^{-2}$. This allowed receptor core side chains to fluctuate and water molecules move freely without significant deviation in the core backbone from its cryo-EM conformation. Prior to restraining the receptor core backbone by 10 kcal mol$^{-1}$ Å$^{-2}$, all systems underwent minimization, heating, and equilibration of the auxiliary proteins as described in the first protocol. Restraints were maintained only on the core backbone throughout the remainder of the trajectories. This protocol was used for the simulations of GluA1/A2$_{Glu+RR2b}$, GluA1/A2-$\gamma8_{Open}$, and GluA1/A2-CNIH2-$\gamma8_{Open}$.

### MD trajectory analysis
Post-processing and analysis of the trajectories were carried out using CPPTRAJ[105] module of AmberTools20 and VMD (1.9.4)[106]. VMD 1.9.4 was used to visualize trajectories and to generate simulation snapshot figures. Water permeation was calculated as the number of water molecules crossing the space between levels T625 and T617 Cα atoms per ns. Cumulative averages were calculated by integrating the mean permeation count over the duration of the simulation. Areas were calculated using the Cα atoms of each gate residue from the four subunits and reported in angstrom (Å) as the square root of actual value.

### Mass spectrometry
Purified protein was resuspended in 100 mM Tris-HCl (pH 8.0), and concentration was determined using the Pierce BCA Protein Assay Kit. A total of 5 μg protein per sample was reduced with 10 mM DTT at 56 °C for 30 min and alkylated with 10 mM iodoacetamide at room temperature in the dark for 30 min, followed by quenching with 5 mM DTT for 15 min. Protein cleanup was performed using SP3 magnetic beads[107], and samples were digested overnight at 37 °C with Lys-C/Trypsin (Promega) at a 1:100 enzyme-to-protein ratio. Digestion was stopped by addition of TFA to 1% (v/v). Peptides were clarified by centrifugation (14,000 × g, 10 min), desalted using SDB-RPS StageTips[108], dried by SpeedVac, and reconstituted in 3% acetonitrile with 0.1% formic acid.

Peptides were separated using a Thermo Scientific UltiMate 3000 RSLCnano system equipped with an EASY-Spray source. Samples were loaded onto an Acclaim PepMap 100 trap column (2 cm × 75 μm) and separated on a C18 analytical column (25 cm × 75 μm, 1.5 μm; IonOpticks) using a 5–35% acetonitrile gradient in 0.1% formic acid over 50 min at 300 nL/min. The column temperature was maintained at 50 °C. Mass spectrometry was performed on an Orbitrap Fusion Tribrid instrument. MS1 scans were acquired from 350–1500 m/z at 120,000 resolution (at 200 m/z), AGC target 3 × 10$^6$, and 50 ms maximum injection time. Data-dependent acquisition was performed in top-speed mode (3 s cycle time), selecting precursors with charge states 2–6 and intensity >5 × 10$^4$ for HCD fragmentation (30% NCE) following quadrupole isolation (1.6 Th). MS2 spectra were acquired in the ion trap at rapid scan rate with dynamic exclusion set to 22 s (10 ppm tolerance). Monoisotopic precursor selection was enabled.

Raw mass spectrometry data were analyzed using MSFragger (v4.3) within FragPipe (v23)[109]. Spectra were searched against the UniProt human database with common contaminants, using a reversed decoy strategy for FDR estimation. Trypsin specificity with up to two missed cleavages was applied. Precursor and fragment mass tolerances were set to 20 ppm. Carbamidomethylation (C) was defined as a fixed modification, with oxidation (M) and protein N-terminal acetylation as variable modifications. Peptide-spectrum matches were filtered to 1% FDR at the PSM and protein levels using PeptideProphet and ProteinProphet. Label-free quantification was performed using IonQuant with match-between-runs enabled.

### Statistics and reproducibility
Statistical analysis (Figs. 1, 6 and Supplementary Fig. 9) was performed using Origin 9.1.0 (OriginLab). Statistical significance was calculated using two-sample $t$-Test, with the significance assumed if $P < 0.05$. In all figure legends, $n$ represents the number of independent biological replicates. All quantitative data were presented as mean ± SEM.

### Reporting summary
Further information on research design is available in the Nature Portfolio Reporting Summary linked to this article.

## Data availability
The cryo-EM density maps have been deposited to the Electron Microscopy Data Bank (EMDB) under the accession codes EMD-70909 (consensus map for GluA1/A2$_{ZK}$), EMD-70912 (composite for GluA1/A2$_{ZK}$), EMD-70911 (LBD of GluA1/A2$_{ZK}$), EMD-70910 (TMD for GluA1/A2$_{ZK}$), EMD-70913 (consensus for GluA1/A2$_{Glu+RR2b}$), EMD-70919 (composite for GluA1/A2$_{Glu+RR2b}$), EMD-75274 (ATD for GluA1/A2$_{Glu+RR2b}$), EMD-70914 (LBD for GluA1/A2$_{Glu+RR2b}$), EMD-70915 (TMD for GluA1/A2$_{Glu+RR2b}$), EMD-70916 (consensus for GluA1/A2-CNIH1$_{Glu+RR2b}$), EMD-70920 (composite for GluA1/A2-CNIH1$_{Glu+RR2b}$), EMD-70917 (LBD for GluA1/A2-CNIH1$_{Glu+RR2b}$), EMD-70918 (TMD for GluA1/A2-CNIH1$_{Glu+RR2b}$), EMD-70921 (consensus for GluA1/A2$_{Quis}$), EMD-70925 (composite for GluA1/A2$_{Quis}$), EMD-70922 (ATD for GluA1/A2$_{Quis}$), EMD-70923 (LBD for GluA1/A2$_{Quis}$), and EMD-70924 (TMD for GluA1/A2$_{Quis}$). The atomic coordinates have been deposited to the Protein Data Bank (PDB) under the accession codes 9OVT (LBD-TMD for GluA1/A2$_{ZK}$), 9OVU (full length for GluA1/A2$_{Glu+RR2b}$), 9OVV (LBD-TMD for GluA1/A2-CNIH1$_{Glu+RR2b}$), and 9OVW (full length for GluA1/A2$_{Quis}$). For illustrations and analysis, we also used structures that correspond to the following PDB entries: 7QHH, 9MRL, 7RZA, 7RZ7, 8SSB, 7QHB, 7OCF, 5KK2, 5WEO, 5VOV, 5VHY, and 5VHZ. The molecular dynamics simulation trajectories and topology files have been deposited to Zenodo [10.5281/zenodo.18627963]. The repository also includes the initial coordinates for the production runs, as well as a table summarizing the corresponding file names. The raw simulation data have not been deposited due to their size, but are available from the authors upon request. The source data underlying Fig. 1c-f are provided as a Source Data file. Source data are provided with this paper.

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

## Acknowledgements

We thank R. Grassucci and Z. Zhang (Columbia University Cryo-Electron Microscopy Center) and G. Angiulli and C. Hernandez (Simons Electron Microscopy Center) for help with microscope operation and data collection. We thank Rajesh Kumar Soni and Caroline W. Karanja (Proteomics and Structural Biology Shared Resource, Herbert Irving Comprehensive Cancer Center, Columbia University, New York, NY) for help with mass spectrometry. Some of this work was performed at the Columbia University Cryo-Electron Microscopy Center. Some of this work was performed at the National Center for CryoEM Access and Training (NCCAT) and the Simons Electron Microscopy Center located at the New York Structural Biology Center, supported by the National Institutes of Health (NIH) Common Fund Transformative High Resolution Cryo-Electron Microscopy program (U24 GM129539) and by grants from the Simons Foundation (SF349247) and NY State Assembly Majority. Some of this work was performed at the Stanford-SLAC Cryo-EM Center (S2C2), supported by the NIH Common Fund Transformative High Resolution Cryo-Electron Microscopy program (U24 GM129541). LYY was supported by NIH grant F31NS132554. TPN was supported by NIH grant F31NS147755. SPG was supported by the NIH grant NS139087. AIS was supported by the Human Frontier Science Program (HFSP) Award and the NIH grants (NS083660, NS107253, AR078814, CA206573).

## Author contributions

L.Y.Y. and A.I.S. conceptualized the project. L.Y.Y., T.P.N., M.V.Y., S.P.G., and A.I.S. designed the experiments. L.Y.Y. made constructs for protein expression and electrophysiology, performed protein expression and purification, made the grids, and collected cryo-EM data. L.Y.Y., T.P.N., and S.P.G. processed the cryo-EM data. M.V.Y. performed patch-clamp recordings and electrophysiological data analysis. M.A. and M.G.K. designed computational studies. M.A. performed MD simulations and MD analysis. L.Y.Y. and A.I.S. built molecular models. RPC synthesized (R,R)–2b. L.Y.Y. and A.I.S. wrote the manuscript, which was then edited by all the authors. AIS supervised the project.

## Competing interests

The authors declare no competing interests.
