## [Transparent Peer Review file · Nature Communications]

Auxiliary subunits reshape structural asymmetry and functional plasticity in heterotetrameric GluA1/A2 AMPA receptor core

Corresponding Author: Dr Alexander Sobolevsky

Version 0:

Reviewer comments:

Reviewer #1

(Remarks to the Author)

This manuscript uses cryo-EM to analyze the structures of the open, closed and desensitized states of heteromeric GluA1/GluA2(R) AMPA receptors, which are likely to be the subunit combination present at the majority of CNS synapses. The work stands out for its thoroughness and the ability to resolve structures within the channel pore that have hitherto been only poorly visualized. In addition to A1/A2 heteromers in isolation, the manuscript also includes interesting information about receptors combined with a variety of auxiliary subunits, including the important observation of a correlation between structural parameters and the rate of recovery from desensitization. Overall the manuscript is well written and the figures illustrate things well.

In my opinion, when describing the molecular dynamic simulations for water flux rates, the text should indicate that the backbone carbons were fixed in place. This information is present in the methods, but might escape the attention of most readers. In this regard the value of Extended Figure 6 was not clear to me. It indicates stability of the pore cross section, measured from C-alpha, throughout the 500 ns runs, but if the backbone was held fixed in place as described in methods wouldn't these C-alpha measurements have to be constant? Why the need to illustrate it with a figure? Clarification of this issue would be welcome in case I've misunderstood either the method or the analysis in this figure.

Reviewer #2

(Remarks to the Author)

This study presents cryo-EM structures of the major forebrain AMPAR, the hetero-tetrameric GluA1/A2 core, in closed, open, and desensitized states, both alone and in complex with the auxiliary subunit CNIH2. These structures provide a crucial baseline for understanding how auxiliary subunits reshape AMPAR gating and functional plasticity. However, several issues in this manuscript remain to be addressed. The suggestions provided here may help the authors enhance the quality of their work to meet the publication criteria of Nature Communications.

Major:

1. There is a marked disparity in sample sizes across the experimental groups in Figure 1e. Why is the sample size (n) for the GluA1/A2 data only 4, while it is 9 for the comparative GluA1/A2 data?
2. Given the similar in vitro purification conditions, what is the molecular mechanism by which quisqualate stabilizes the complex in the desensitized state? Can potential mechanisms be proposed based on previous structural studies or the structures resolved in this work?
3. The study uses rat GluA1/A2 without overexpressing auxiliary subunits such as CNIH2. It is reasonable to assume that these subunits originate from the human HEK293 expression system. Could sequence differences between species affect the results? Does this pattern observed truly represent the binding mode in a homogeneous system?
4. At line 240, why was an unrestrained simulation used for the MD of GluA1/A2ZK, while backbone restrained simulations were applied in other cases?

5. What is the molecular mechanism by which CNIH2 regulates the core amplification of GluA1/A2? Does this allosteric regulatory mechanism present potential for drug design?
6. The MD data exhibit high heterogeneity, and results from a single simulation may be coincidental. It is recommended that the MD experiments be repeated in over three independent replicates to ensure the robustness of the conclusions.
7. In Figure 5e, if statistics are derived from a single MD simulation trajectory, what is the total amount of data used for error calculation? Do the error bars represent SEM or SD? If the data are averaged from multiple independent MD runs, SEM is recommended. If the plot represents a time trajectory from a single simulation, SD is more appropriate.
8. The refinement statistics, including the Ramachandran plot and the percentage of poor rotamers, are suboptimal. The structural model should be optimized to achieve a Ramachandran plot with >93% favored residues (ideally >95%) and <0.5% disallowed residues (ideally 0%). Poor rotamers should also be minimized. For regions with poor density in the GluA1/A2 quis structure, consider removing most side chains while ensuring correct backbone placement.
9. Please include the Model resolution (FSC threshold at 0.5) in the Refinement section of the Extended Data Tables. This parameter, which can typically be found in validation reports from software such as Phenix, assesses the overall fit of the model coordinates to the map and should be slightly higher than the Map resolution (FSC threshold at 0.143).

Minor:

1. Certain specialized terms (e.g., "D1" and "D2 lobes") should be briefly explained upon their first occurrence.
2. In describing the tetrameric conformation, the authors refer to GluA1 and GluA2 using "A/B/C/D positions." Is this nomenclature specific to AMPAR? If so, relevant references should be provided. Otherwise, a brief explanation at Line 151 is recommended.
3. What is the structural basis for identifying the quisqualate-bound GluA1/A2 quis structure as a desensitized state? It is advised to align this structure with previously resolved ones and provide supporting evidence.
4. How does the conformation of D1/D2 compare with previously resolved AMPAR structures? The description in lines 179–187 could be better illustrated with a figure. If these structural features are similar to known structures, what are the unique characteristics of the structures resolved in this study?
5. At line 988, it is suggested to add labels "a–e" at the beginning.
6. The label in the figures (e.g., "GluA1," "GluA2" labels in Figure 3a) is hard to distinguish and should be refined.
7. How can the abrupt drop and subsequent recovery observed in the FSC curves (e.g., in Extended Data Figure 3, the FSC for TMD GluA1/A2 quis approaches 0.143 near 6 Å) be explained?
8. Figure 6e and Extended Data Figure 7a,b lack indications of the number of independent biological replicates. It is also recommended to follow Nature communications' figure guidelines (e.g., Figure 1c-f) by including individual data points in bar graph.
9. Remove "(R)" at line 109, as this abbreviation was already introduced at line 66.

Reviewer #3

(Remarks to the Author)

The manuscript by Laura Y. Yen et al, presents structures of a GluA1/A2 AMPA receptor in multiple functional states (open, resting, and desensitized conformations), as well as a complex with a cornichon-like homologue natively expressed in HEK cells. The availability of a GluA1/A2 reference structure is valuable for understanding TARP modulation. Notably, this manuscript reports the first open state of an AMPA receptor without auxiliary subunits at side-chain resolution, as well as the first open structure of an AMPAR:CNIH complex, which represent a significant advance in AMPAR structure-function studies.

However, I have several major concerns that should be addressed. My primary concern relates to the use of modified constructs for both GluA1 and GluA2, specifically the shortened ATD–LBD linkers. While these modifications used to be necessary for crystallization, multiple recent cryo-EM studies have successfully determined structures of wild-type AMPA receptors, including GluA1/A2 complexes with TARP8 (\pm CNIH2). Moreover, the authors' own data indicate that the construct used here exhibits slower desensitization kinetics, resembling TARP-bound complexes more than TARP-free AMPARs. As a result, the construct may not represent a physiologically accurate GluA1/A2 receptor, but rather a receptor with reduced conformational flexibility and altered desensitization behavior. This raises concerns about its suitability as a reference model. Further, previous work has shown that shortening these linkers accelerates recovery from desensitization in AMPAR–TARP complexes (Cais et al., Cell Reports, 2014). The only cryo-EM structure of GluA2 alone with wild-type linkers displayed substantial conformational variability in the desensitized state (Meyerson et al., Nature 2014), a feature not observed in the present work. Thus, it remains insufficiently justified how the modified construct reported here can serve as a

reference for understanding GluA1/A2 desensitization mechanisms. If the authors intend to draw conclusions regarding entry into, or recovery from, desensitization, a wild-type structure is essential.

In addition, the authors indicate that the ATD does not contribute to receptor functionalities (line 153). However, recent studies (Ivica et al., 2024, NSMB; Larsen et al., 2024, JBC; Zhang et al., 2023, Nature) demonstrate that the ATD layer (and specifically, its stable tetrameric interface), plays a fundamental regulatory role in recovery from desensitization. It is surprising that the ATD was only clearly defined in the desensitized state but not in open and resting conditions, when the opposite has been suggested in receptors with WT linkers. The linker mutations employed here may reduce the ATD mobility and may consequently mask functionally important transitions. This limitation would be mitigated if the authors were to provide a control structure of GluA1/A2 with fully wild-type linkers in a desensitized state. While such a structure may exhibit greater flexibility and lower resolution, it would be critical for validating the mechanistic interpretations presented and for supporting claims regarding desensitization properties. Moreover, current cryo-EM approaches do permit analysing dynamics, and this has been done in receptors which much more flexibility than GluA2, such as GluA1, GluA3 and GluA4. Such analysis should also be shown here. It would indeed be essential to confirm subunit positioning (as key differences between subunits are located at the ATD) as well as to discard the absence of cis-AMPA receptors (which have been observed in GluA1 receptors).

The second major concern relates to the identification of CNIH2 in the open-state dataset. Do the authors have unequivocal evidence that CNIH2 is expressed in HEK cells? Considering RNA expression databases, CNIH2 expression is low in HEK cells, whereas other cornichon homologues (notably CNIH1) are expressed at high levels. Can the authors directly trace CNIH2-specific side chains in the EM density to confirm the identity of the auxiliary subunit? qPCR or mass spectrometry data demonstrating CNIH2 expression would be required to support this assignment. Alternatively, electrophysiological characterization of the BacMam-expressed receptor complex—showing functional hallmarks of CNIH2 modulation—would help to validate the conclusion.

Finally, several aspects of the cryo-EM processing and data analysis require clarification. These include corrections to models and tables (for example, Table 2 reports 1.82% Ramachandran outliers in one model, which is unexpected at the reported resolution and should be revised). In addition, local-resolution maps appear to show unusual features (e.g., the TMD map in Extended Data Fig. 1, or the first LBD map in Extended Data Fig. 2, where the map seem to display substantially higher resolution than stated considering FSC graph). The authors should carefully re-examine these maps and provide a clearer description of “composite” and consensus maps, including how composite-map resolution was estimated and why particle numbers differ between open-state consensus, composite, and focused maps. No Ramachandran outliers are expected.

Other comments:

- Line 65 “For instance, GluA2-containing AMPARs are generally Ca²⁺-impermeable due to RNA editing that substitutes 65 glutamine (Q) for arginine (R) at position 586, the Q/R site, within the channel pore. The distinction between Ca²⁺-permeable and Ca²⁺-impermeable AMPARs should be revisited in light of recent publications (Miguez-Cabello et al., Nature, 2025).

-Line 74 “all available structures of the homomeric core alone have suffered from both low resolution and absence of the critical re-entrant M2 loop of the transmembrane domain (TMD), which lines the pore in conjunction with helix M3 and forms the channel’s selectivity filter”

This statement is not accurate. The structure of wild-type GluA4 without auxiliary proteins was recently determined at side-chain resolution and showed clear density for the M2 region (Vega-Gutierrez et al., NSMB, 2025).

-Line 151: “Due to the flexibility of the ATD layer that worsens the overall resolution of the full length receptor map and as it does not contribute to AMPAR functionalities, we focused our structural analysis on the LBD-TMD region”.

This assertion is not precise. Multiple recent studies (Ivica et al., NSMB 2024; Larsen et al., JBC 2024; Zhang et al., Nature 2023) demonstrate that the ATD (specifically the GluA2 ATD tetramer interface), plays a critical role in recovery from desensitization. While it is true that the ATD is flexible, current cryo-EM methods can reliably capture this variability, especially in GluA2-containing receptors, where the ATD dimer interface is stable. Furthermore, having the ATD structure is essential to correctly assign subunits within the tetramer. Given the high sequence conservation in the LBD and TMD, the manuscript does not presently provide sufficient evidence for the subunit positioning reported. Therefore, the analysis of the NTD layer should also be presented. Have the authors identified unambiguous, subunit-specific residues in the density maps corresponding to GluA2 or GluA1?

- Figure 6:

Panel (e) contains incorrect references: reference 65 is a proteomic study and does not report GluA1/A2/TARP8 functional characterization. Additionally, comparing recovery data across whole-cell and outside-out patch recordings is problematic, as major differences in desensitization (both entry and recovery) have been shown in back-to-back experiments (Zhang et al., Nature, 2023). The table also mixes data from constructs with shortened linkers and constructs with wild-type linkers, even though linker length is known to affect recovery kinetics in AMPAR–TARP complexes. Considering this concerns as well as the fact that most data in panel (e) have not been generated in this manuscript, it may be better to exclude it from a main text figure.

- Extended Data Figure 1:

Please check the handedness of the grey map in the classification step, it looks like it is inverted. Also Re-examine the local-resolution calculations for the TMD map; the left side displays an unusual blue pattern that should be verified. Please complete the figure legend.

-Cryo-EM analysis of the open state:

Please, describe with a figure the 3D processing strategy for the open state. Currently, the number of particles in Table 1 and 2 do not match the processing strategy defined in the text (131,148 particles vs 151,693 -LBD, 74183 for the TMD and 61870 particles): has the composite map been obtained with two different sets of particles? Please clarify this. In general Table 1 and Table 2 and methods are not very clear: for example, how is the resolution of the composite map calculated? For the CNIH complex, both composite and consensus maps are reported, but the focused maps used to build the composite map are not documented. Composite maps should be reserved for visualization only, particularly when derived from different particle subsets.

-Cryo-EM density details:

Please include CNIH2 density in Extended Data Figure 4, especially for regions that distinguish CNIH2 from CNIH1. Also provide density features confirming GluA1 vs. GluA2 subunit identity (e.g., the Q/R site, R/G site, or other subunit-specific residues that support the assignment).

-Extended Data Fig. 7:

Correct the reference for the GluA1/A2/TARP8 dataset and clearly specify differences in recording configuration (whole-cell vs. outside-out) and in constructs used (wild type vs. modified).

Version 1:

Reviewer comments:

Reviewer #1

(Remarks to the Author)

The authors have addressed my concerns and made extensive revisions to their manuscript in response to all three of the initial reviews. The important advances in this study were noted in the initial reviews and the revised manuscript is much improved. No additional concerns were noted.

Reviewer #2

(Remarks to the Author)

The authors have addressed my concerns.

Reviewer #3

(Remarks to the Author)

Thank you very much for the revised version of the manuscript.

The authors have addressed most of my comments. I still think that the structural work presented here would be more relevant if WT constructs were used, as current GluA1/A2 structures in complex with auxiliary proteins are WT. Still, the manuscript presents very useful data, including CNIH1 binding and open states without auxiliary proteins. The MS data, as well as the details of the cryo-EM maps in the CNIH TMD, clearly support the binding of CNIH1 to AMPARs in HEK cells. This opens new questions about a possible role of CNIH1 in trafficking or kinetic modulation.

Please add the mass spectrometry methods.

I do not have any more comments.

Thanks,

We thank all Reviewers for their excellent suggestions that have led to significant improvement of this manuscript. We have made changes throughout the manuscript with the details outlined in our responses below.

Reviewer #1 (Remarks to the Author):

This manuscript uses cryo-EM to analyze the structures of the open, closed and desensitized states of heteromeric GluA1/GluA2(R) AMPA receptors, which are likely to be the subunit combination present at the majority of CNS synapses. The work stands out for its thoroughness and the ability to resolve structures within the channel pore that have hitherto been only poorly visualized. In addition to A1/A2 heteromers in isolation, the manuscript also includes interesting information about receptors combined with a variety of auxiliary subunits, including the important observation of a correlation between structural parameters and the rate of recovery from desensitization. Overall the manuscript is well written and the figures illustrate things well.

We thank Reviewer #1 for the positive and generous assessment of our work.

In my opinion, when describing the molecular dynamic simulations for water flux rates, the text should indicate that the backbone carbons were fixed in place. This information is present in the methods, but might escape the attention of most readers. In this regard the value of Extended Figure 6 was not clear to me. It indicates stability of the pore cross section, measured from C-alpha, throughout the 500 ns runs, but if the backbone was held fixed in place as described in methods wouldn't these C-alpha measurements have to be constant? Why the need to illustrate it with a figure? Clarification of this issue would be welcome in case I've misunderstood either the method or the analysis in this figure.

We thank the reviewer for thoughtful comments and for the opportunity to clarify the methodology.

- 1) We agree that it is important to clearly state in the main text that the simulations for water flux rate were performed with backbone atoms restrained. As the reviewer notes, this information is provided in the Methods, and we already refer to these simulations in the text (line 267) as "backbone-restrained simulations" of the cryo-EM structures. This description refers to restraints applied to all backbone heavy atoms (N, C α , C, and O), not only the backbone carbons. To avoid any ambiguity, we revised the text to explicitly state this earlier in the Results section (line 268-270).
- 2) The time-series plot of cross-sectional areas in the former **Extended Figure 6** was initially added to emphasize that the gate areas do not change in backbone-restrained simulations. However, we agree with the reviewer, and we have therefore replaced the figure with a bar plot showing the mean cross-sectional gate areas for each simulation (**Supplementary Figure 8**). We believe this new presentation more clearly communicates the relevant information without implying unnecessary time-dependent fluctuations.

Reviewer #2 (Remarks to the Author):

This study presents cryo-EM structures of the major forebrain AMPAR, the hetero-tetrameric GluA1/A2 core, in closed, open, and desensitized states, both alone and in complex with the auxiliary subunit CNIH2. These structures provide a crucial baseline for understanding how auxiliary subunits reshape AMPAR gating and functional plasticity. However, several issues in this manuscript remain to be addressed. The suggestions provided here may help the authors enhance the quality of their work to meet the publication criteria of Nature Communications.

We thank Reviewer #2 for the kind assessment of our work.

Major:

1. There is a marked disparity in sample sizes across the experimental groups in Figure 1e. Why is the sample size (n) for the GluA1/A2 data only 4, while it is 9 for the comparative GluA1/A2 data?

We carried out additional experiments, increased the number of measurements and updated **Fig. 1** accordingly.

2. Given the similar in vitro purification conditions, what is the molecular mechanism by which quisqualate stabilizes the complex in the desensitized state? Can potential mechanisms be proposed based on previous structural studies or the structures resolved in this work?

The mechanism of desensitization induced by quisqualate appears to be similar to the one proposed before (doi: 10.1038/417245a; doi: 10.1016/j.cell.2017.07.045; doi: 10.1016/j.neuron.2017.04.025). According to this mechanism, quisqualate binding causes LBD clamshell closure. However, in contrast to receptor activation, during which the D1-D1 interface in LBD dimer remains intact, the LBD closure during desensitization causes the D1-D1 interface rupture. As a result, the D1 lobes undergo separation, while the D2 lobes remain close to each other, like in the closed state. The lack of D2-D2 lobe separation, which drives pore dilation in the open state, ensures closed conformation of the ion channel, a characteristic of the desensitized state. We have illustrated this mechanism with examples of LBD behavior originally reported in the full-length AMPARs (new **Supplementary Figure 9a-f**). More detailed description of this mechanism has also been added to the text (lines 293-303).

3. The study uses rat GluA1/A2 without overexpressing auxiliary subunits such as CNIH2. It is reasonable to assume that these subunits originate from the human HEK293 expression system. Could sequence differences between species affect the results? Does this pattern

observed truly represent the binding mode in a homogeneous system?

Since receiving the reviewer comments, we've performed some follow-up experiments. Specifically, we've run mass spectrometry on the GluA1/A2 purified protein to identify the identity of co-purified CNIH. We find that all 4 paralogs (CNIH1, CNIH2, CNIH3 and CNIH4) are present in the sample, with strongest evidence suggesting the major co-purified species and CNIH1. As such, we've provided sequence alignment for human CNIH1-4, as well as rat CNIH1-4

```

hCNIH1      -----MAFTFAAFCYMLALLLTAALIFFAIWHIIAFDELK  35
rCNIH1      -----MAFTFAAFCYMLALLLTAALIFFAIWHIIAFDELK  35
hCNIH2      -----MAFTFAAFCYMLTLVLCASLIFFVIWHIIAFDELR  35
rCNIH2      VWAGGRPLAPRPRAGAPRATARRRGGHGVHLRSIL-LHAHPGAVRLPHLLCHLIIAFDELR  59
hCNIH3      -----MAFTFAAFCYMLSLVLCALIFFAIWHIIAFDELR  35
rCNIH3      -----MAFTFAAFCYMLSLVLCALIFFAIWHIIAFDELR  35
hCNIH4      -----MEAVVVFVSLLDCCALIFLSVYFIITLSDLE  31
rCNIH4      -----MEAVVFLFSLLDCCSLIFLSVYFIITLSDLE  31
                                     : . : : * . : **:::*

hCNIH1      TDYKNPIDQCNTLNP-----LVLPEYLIHAFFCVMFLCAAEWLTLGLN  78
rCNIH1      TDYKNPIDQCNTLNPTVEKVKIKRV-KIALKLVLPEYLIHAFFCVMFLCAAEWLTLGLN  94
hCNIH2      TDFKNPIDQGNPARAR-ERLKNIERICLLRKLVVPEYSIHGLFCLMFLCAAEWTLGLN  94
rCNIH2      TDFKNPIDQGNPARAR-ERLKNIERICLLRKLVVPEYSIHGLFCLMFLCAAEWTLGLN  118
hCNIH3      TDFKSPIDQCNPVHAR-ERLRNIERICFLLRKLVLPEYSIHSLFCIMFLCAQEWLTLGLN  94
rCNIH3      TDFKSPIDQCNPVHAR-ERLRNIERICFLLRKLVLPEYSIHSLFCIMFLCAQEWLTLGLN  94
hCNIH4      CDYINARSC-----CSKLNKWWIPELIGHTIVTVLLLSLHWFIFLLN  74
rCNIH4      CDYINARSC-----CSKLNKWWIPELVGHTFVTVLMLVSLHWFIFLLN  74
          * : . .          * : ** * . : : : * : . * : **

hCNIH1      MPLLAYHIWRYMSRPVMSGPLYDPTT IMNADILAYCQKEGWCKLAFYLLAFFYYLYGMI  138
rCNIH1      MPLLAYHIWRYMSRPVMSGPLYDPTT IMNADILAYCQKEGWCKLAFYLLAFFYYLYGMI  154
hCNIH2      IPLLFFYHLWRYFHRPADGSEVMYDAVSIMNADILNYCQKESWCKLAFYLLSFFYYLYSMV  154
rCNIH2      IPLLFFYHLWRYFHRPADGSEVMYDAVSIMNADILNYCQKESWCKLAFYLLSFFYYLYSMV  178
hCNIH3      VPLLFFYHFWRYFHCPADSSELAYDPPVVMNADTLSYCQKEAWCKLAFYLLSFFYYLYCMI  154
rCNIH3      VPLLFFYHFWRYFHCPADSSELAYDPPVVMNADTLSYCQKEAWCKLAFYLLSFFYYLYCMI  154
hCNIH4      LPVATWNIYRFIMVPS-GNMGVFDPTTEIHNRGQLKSHMKEAMIKLGFHLLCFFMYLYSMI  133
rCNIH4      LPVATWNIYRFIMVPS-GNMGVFDPTTEIHNRGQLKSHMKEAMIKLGFYLLCFFMYLYSMI  133
          **:  ::::*: * .. : * : * . * ** . **.*:*** ** ** *:

hCNIH1      YVLVSS  144
rCNIH1      YVLVSS  160
hCNIH2      YTLVSF  160
rCNIH2      YTLVSF  184
hCNIH3      YTLVSS  160
rCNIH3      YTLVSS  160
hCNIH4      LALIND  139
rCNIH4      LALIND  139
          .*:

```

Given the same binding interfaces and positioning of cornichon proteins in all previously reported AMPAR structures¹⁻³, we believe that the reported structure is a fair representation of cornichon binding in natural conditions.

4. At line 240, why was an unrestrained simulation used for the MD of GluA1/A2ZK, while backbone restrained simulations were applied in other cases?

We first started simulating all the structures without any restraints. In these simulations, the structures GluA1/A2- $\gamma 8_{\text{Open}}$ and GluA1/A2-CNIH2- $\gamma 8_{\text{Open}}$ were stable, and their gates remained open throughout the trajectories. Antagonist-bound GluA1/A2_{ZK}, which is a closed-channel structure, was also stable and its gate remained continuously closed during the simulations. However, the gates of GluA1/A2_{Glu+RR2b} and GluA1/A2-CNIH_{Glu+RR2b} structures never remained open in unrestrained simulations. So, we decided to simulate all the structures with their backbone atoms restrained to be able to compare the water conductances in conformations closest to their cryo-EM structures. Since the simulation of GluA1/A2_{ZK} was very stable and in non-conducting state, we decided not to run an additional backbone-restrained simulation of this structure.

5. What is the molecular mechanism by which CNIH2 regulates the core amplification of GluA1/A2? Does this allosteric regulatory mechanism present potential for drug design?

Since CNIH proteins only have the transmembrane part and do not have extracellular domains, like TARPs, their allosteric mechanism of action likely relies on the alteration of the AMPAR transmembrane domain architecture and dynamics, which are critical for the receptor gating. Accordingly, small molecules perturbing AMPAR-CNIH interfaces have a potential to become drugs. In this regard, a perspective location for drug targeting is the extracellular part of the AMPAR-CNIH interface, which is adjacent to the AMPAR extracellular collar involved in gating and hosting anti-epileptic drugs like perampanel^{1,4,5}. For instance, a similar extracellular side membrane interface in AMPAR-TARPy8 complex has already been targeted with several small molecules⁶⁻⁸. The corresponding information has been added to the text (lines 280-288).

6. The MD data exhibit high heterogeneity, and results from a single simulation may be coincidental. It is recommended that the MD experiments be repeated in over three independent replicates to ensure the robustness of the conclusions.

We appreciate the reviewer's concern regarding heterogeneity and the common recommendation of performing three or more independent MD replicates. In our case, however, the simulations were performed with harmonic restraints applied to all backbone heavy atoms during the production phase, which substantially reduces conformational variability and limits the accessible phase space of the system. As a result, the trajectories are highly reproducible and would show minimal divergence when simulations are initiated from the same equilibrated structure. Under such restrained conditions, multiple replicates are not expected to yield statistically independent outcomes but instead converge to nearly identical behavior, providing little additional information while significantly increasing computational cost. Consequently, we

think that running more than one simulation would not meaningfully change the estimated conductance or strengthen the robustness of the conclusions, since the methodological design itself enforces reproducibility.

7. In Figure 5e, if statistics are derived from a single MD simulation trajectory, what is the total amount of data used for error calculation? Do the error bars represent SEM or SD? If the data are averaged from multiple independent MD runs, SEM is recommended. If the plot represents a time trajectory from a single simulation, SD is more appropriate.

The statistics for each simulation system were derived from a single MD simulation trajectory, with the total simulation time of ~500 ns (~5000 data points) used to calculate water conductance and its variability. We computed the cumulative average water conductance rates by integrating the mean permeation count over the duration of the simulation. Error bars represent standard deviation of water permeation counts within each simulation, shown on the cumulative average bar plot. In response to the reviewer's comments, we have now revised **Figure 5e** to display the water conductance plot with these error bars.

8. The refinement statistics, including the Ramachandran plot and the percentage of poor rotamers, are suboptimal. The structural model should be optimized to achieve a Ramachandran plot with >93% favored residues (ideally >95%) and <0.5% disallowed residues (ideally 0%). Poor rotamers should also be minimized. For regions with poor density in the GluA1/A2quis structure, consider removing most side chains while ensuring correct backbone placement.

We have optimized the GluA1/A2quis model, which now has better Ramachandran statistics (see **Supplementary Table 1**).

9. Please include the Model resolution (FSC threshold at 0.5) in the Refinement section of the Extended Data Tables. This parameter, which can typically be found in validation reports from software such as Phenix, assesses the overall fit of the model coordinates to the map and should be slightly higher than the Map resolution (FSC threshold at 0.143).

The requested information has been added to the Refinement section of **Supplementary Table 1**.

Minor:

1. Certain specialized terms (e.g., "D1" and "D2 lobes") should be briefly explained upon their first occurrence.

Done (lines 168 - 170).

2. In describing the tetrameric conformation, the authors refer to GluA1 and GluA2 using "A/B/C/D positions." Is this nomenclature specific to AMPAR? If so, relevant references should be provided. Otherwise, a brief explanation at Line 151 is recommended.

We have added an explanation of this nomenclature (line 152 – 159) and referenced the original paper⁹.

3. What is the structural basis for identifying the quisqualate-bound GluA1/A2 structure as a desensitized state? It is advised to align this structure with previously resolved ones and provide supporting evidence.

Structural features and the mechanism of desensitization in AMPARs have been proposed before^{10–12}. According to this mechanism, agonist binding causes LBD clamshell closure. However, in contrast to receptor activation, during which the D1-D1 interface in LBD dimer remains intact, the LBD closure during desensitization causes the D1-D1 interface rupture. As a result, the D1 lobes undergo separation, while the D2 lobes remain close to each other, like in the closed state. The lack of D2-D2 lobe separation, which drives pore dilation in the open state, ensures closed conformation of the ion channel, a characteristic of the desensitized state. The quisqualate-bound GluA1/A2 structure, therefore, has all the features of the desensitized state. We have illustrated the desensitization mechanism with examples of LBD behavior originally reported in the full-length AMPARs (new **Supplementary Fig. 9a-f**). We have also added a more detailed description of the desensitization mechanism to the text (lines 293 - 303).

4. How does the conformation of D1/D2 compare with previously resolved AMPAR structures? The description in lines 179–187 could be better illustrated with a figure. If these structural features are similar to known structures, what are the unique characteristics of the structures resolved in this study?

Structural features and the mechanism of activation have been originally proposed in studies of homotetrameric AMPARs^{10,12,13}. We have illustrated the conformational behavior of D1/D2 with examples of LBD behavior originally reported in the full-length homotetrameric AMPARs in the new **Supplementary Fig. 5**. The corresponding explanation has also been added to the text (lines 193 - 199).

5. At line 988, it is suggested to add labels "a–e" at the beginning.

Done.

6. The label in the figures (e.g., "GluA1," "GluA2" labels in Figure 3a) is hard to distinguish and should be refined.

We have updated the labels.

7. How can the abrupt drop and subsequent recovery observed in the FSC curves (e.g., in Extended Data Figure 3, the FSC for TMD GluA1/A2quis approaches 0.143 near 6 Å) be explained?

This drop in the FSC curve is typical of membrane proteins and is a feature present due to phase randomization, which is implemented in all refinement jobs in CryoSPARC¹⁴. In part, the dip in FSC could also be a function of preferred orientation, which is visible in the angular distribution heatmap (strong equatorial enrichment). This is especially common around 4-6 Å resolution, where secondary-structure features dominate and directional information matters most¹⁵.

8. Figure 6e and Extended Data Figure 7a,b lack indications of the number of independent biological replicates. It is also recommended to follow Nature communications' figure guidelines (e.g., Figure 1c-f) by including individual data points in bar graph.

Most of the data presented in **Fig. 6e** and former **Extended Data Fig. 7a-b** are from the previously published work, and we do not have access to individual measurements. To comply with the guidelines, we replaced the bar graphs with symbol plots and introduced the corresponding information (references, SEM or SD, the number of biological replicates) about the data sources in the figure legends.

9. Remove "(R)" at line 109, as this abbreviation was already introduced at line 66.

Done (lines 108-109).

Reviewer #3 (Remarks to the Author):

The manuscript by Laura Y. Yen et al, presents structures of a GluA1/A2 AMPA receptor in multiple functional states (open, resting, and desensitized conformations), as well as a complex with a cornichon-like homologue natively expressed in HEK cells. The availability of a GluA1/A2 reference structure is valuable for understanding TARP modulation. Notably, this manuscript reports the first open state of an AMPA receptor without auxiliary subunits at side-chain resolution, as well as the first open structure of an AMPAR:CNIH complex, which represent a

significant advance in AMPAR structure-function studies.

We thank Reviewer #3 for the generous assessment of our work.

However, I have several major concerns that should be addressed. My primary concern relates to the use of modified constructs for both GluA1 and GluA2, specifically the shortened ATD–LBD linkers. While these modifications used to be necessary for crystallization, multiple recent cryo-EM studies have successfully determined structures of wild-type AMPA receptors, including GluA1/A2 complexes with TARP8 (\pm CNIH2). Moreover, the authors' own data indicate that the construct used here exhibits slower desensitization kinetics, resembling TARP-bound complexes more than TARP-free AMPARs. As a result, the construct may not represent a physiologically accurate GluA1/A2 receptor, but rather a receptor with reduced conformational flexibility and altered desensitization behavior. This raises concerns about its suitability as a reference model. Further, previous work has shown that shortening these linkers accelerates recovery from desensitization in AMPAR–TARP complexes (Cais et al., Cell Reports, 2014). The only cryo-EM structure of GluA2 alone with wild-type linkers displayed substantial conformational variability in the desensitized state (Meyerson et al., Nature 2014), a feature not observed in the present work. Thus, it remains insufficiently justified how the modified construct reported here can serve as a reference for understanding GluA1/A2 desensitization mechanisms. If the authors intend to draw conclusions regarding entry into, or recovery from, desensitization, a wild-type structure is essential.

The difference mentioned for the recovery from desensitization from Cais et al (2014)¹⁶ has only been shown for GluA2i-y2 and GluA2i-y8 (the latter one, even though statistically significant, but numerically very similar values). For the GluA2i receptor alone and GluA2i-y3 and GluA2i-y4, there was no difference between WT and shortened ATD–LBD linker constructs. Similarly, in our case, the rate of recovery from desensitization for GluA1/A2 does not depend on the linker length (**Fig. 1e**). Moreover, the fraction of non-desensitized channels (**Fig. 1f**) and the time constants of deactivation (**Fig. 1d**) for WT and shortened ATD–LBD linker GluA1/A2 are also similar, suggesting that our structures are fairly accurate representations of physiological receptors. We agree with Reviewer #3 that 1.8-times slower rate of desensitization in shortened ATD–LBD linker GluA1/A2 compared to WT (**Fig. 1c**) may indicate a possible influence of the length of the ATD–LBD linker on the process of entry into desensitization, but given the lack of effect on the fraction of non-desensitized channels (**Fig. 1f**) and the rate of recovery from desensitization (**Fig. 1e**), this small (from the biophysical perspective) difference in the rate of desensitization is unlikely to result in dramatic changes in the desensitized state structure. Nevertheless, when describing the desensitized state structure of GluA1/A2, we have now mentioned the above limitations (lines 129-133). Besides, Reviewer #3 is incorrect that our desensitized state structure lacks substantial conformational variability of the desensitized state observed in the structure of Meyerson et al. (Nature 2014)¹⁷. Our 2D class averages include plenty of images with clear blurriness of the ATD layer relative to LBD-TMD (or vice versa). It is simply because good computational tools that allow focused classification and refinement have

not been available in 2014, Meyerson et al. were not able to get good 3D reconstruction. In addition, the quality of Meyerson et al. data was substantially worse than ours. Given all of the above, we do not consider the structure of WT GluA1/A2 as necessary for the present manuscript, specifically noting the timeframe for this project exceeded ~5 years to optimize the current constructs, expression and sample preparation and it might easily take the same time or more for the WT channels, while the benefit of such structures compared to the presented ones in this manuscript is questionable.

In addition, the authors indicate that the ATD does not contribute to receptor functionalities (line 153). However, recent studies (Ivica et al., 2024, NSMB; Larsen et al., 2024, JBC; Zhang et al., 2023, Nature) demonstrate that the ATD layer (and specifically, its stable tetrameric interface), plays a fundamental regulatory role in recovery from desensitization. It is surprising that the ATD was only clearly defined in the desensitized state but not in open and resting conditions, when the opposite has been suggested in receptors with WT linkers. The linker mutations employed here may reduce the ATD mobility and may consequently mask functionally important transitions. This limitation would be mitigated if the authors were to provide a control structure of GluA1/A2 with fully wild-type linkers in a desensitized state. While such a structure may exhibit greater flexibility and lower resolution, it would be critical for validating the mechanistic interpretations presented and for supporting claims regarding desensitization properties. Moreover, current cryo-EM approaches do permit analysing dynamics, and this has been done in receptors which much more flexibility than GluA2, such as GluA1, GluA3 and GluA4. Such analysis should also be shown here. It would indeed be essential to confirm subunit positioning (as key differences between subunits are located at the ATD) as well as to discard the absence of cis-AMPA receptors (which have been observed in GluA1 receptors).

Given the mentioned examples^{6,18,19}, we have corrected the sentence on line 153 (now line 165). Due to the rigid body motion of the entire ATD layer relative to LBD layer, it is impossible to get a good consensus reconstruction of the entire FL receptor, demonstrated by consistent fuzziness of the ATD layer in 2D class average during data processing. Because of limited influence of ATD on function of AMPA receptors with GluA1/A2 core in general, it has become a common practice in the field of AMPA receptor structural biology to mainly focus on LBD-TMD and ignore ATD^{3,20-22}. Nevertheless, we decided to illustrate the rigid-body assembly of the ATD layer on the example of the desensitized state to contrast it to the behavior observed before for some AMPARs, in which ATD dimers were reported to dissociate from each other during desensitization^{17,23}. To further illustrate the same rigid-body behavior of ATD, we have now added a similar reconstruction of the ATD layer for the open state (**Supplementary Fig. 2**). We also do not expect the rigid-body “floating” of ATD layer above the LBD layer to change with increased length of the ATD-LBD linkers, except that the amplitude of the motion increases and blurriness of the ATD layer increases correspondingly. Given the limited effect of the length of the ATD-LBD linkers of GluA1/A2 function (**Fig. 1**) and the difficulty in getting the FL structures (see the previous point), we consider determination of the desensitized-state structure of WT GluA1/A2 to be outside the scope of the present manuscript. Since we now report excellent densities for the ATD layer, we have included illustrations of cryo-EM density for the

carbohydrates at the predicted ATD sites that uniquely distinguish subunits GluA1 and GluA2 and unambiguously verify the established order of subunits in the GluA1/A2 tetramer (**Supplementary Fig. 4g-j**), consistent with the previous reports^{3,20,22}. Furthermore, with our optimized expression and purification conditions, dual-step affinity chromatography significantly depletes the possibility for homomeric AMPARs to be purified. Specifically, covalently linking a His-tag to our GluA1 construct allows for the removal of homomeric GluA1 during the first affinity chromatography step using streptavidin-linked affinity purification, leaving only homomeric GluA2 and heteromeric GluA1/A2. Then, a second step of nickel-based affinity chromatography to bind His-tags on GluA1 removes homomeric GluA2. Therefore, the remaining tetrameric AMPARs present in our sample are GluA1/A2 receptors only.

The second major concern relates to the identification of CNIH2 in the open-state dataset. Do the authors have unequivocal evidence that CNIH2 is expressed in HEK cells? Considering RNA expression databases, CNIH2 expression is low in HEK cells, whereas other cornichon homologues (notably CNIH1) are expressed at high levels. Can the authors directly trace CNIH2-specific side chains in the EM density to confirm the identity of the auxiliary subunit? qPCR or mass spectrometry data demonstrating CNIH2 expression would be required to support this assignment. Alternatively, electrophysiological characterization of the BacMam-expressed receptor complex—showing functional hallmarks of CNIH2 modulation—would help to validate the conclusion.

We are very grateful to Reviewer #3 for this comment. Previously, we blindly trusted the published papers that stated complete lack of interaction between AMPARs and CNIH1/4^{24–26}. We have now submitted our purified protein to mass spectrometry and found that we have a mixture of all four cornichons, CNIH1-4, in our sample (**Supplementary Table 2**). The strongest signal was observed for CNIH4, followed by CNIH1, CNIH3 and CNIH2. We think that all four CNIHs are likely to represent the corresponding cryo-EM density. Despite CNIH4 is the most abundant CNIH in our sample according to mass spectrometry, there is cryo-EM density for the N-terminal end of CNIH, which is present in CNIH1-3 but not in CNIH4 (**Supplementary Fig. 6f-j**), pointing to CNIH1-3 as necessary contributors to the observed density. The rest of the cryo-EM density fits all three remaining CNIHs, CNIH1-3, almost equally well, which is understandable given high sequence similarity. In a few locations illustrated in **Supplementary Fig. 6k-t**, cryo-EM density appears to match CNIH1 somewhat better than CNIH 2 and CNIH3. We therefore decided to illustrate CNIH binding to GluA1/A2 by modeling the auxiliary protein density with CNIH1 and use this model for illustrations. However, given that all four CNIHs are likely to contribute to the average density, we cautiously call the auxiliary subunits in our illustrations as CNIHs and have added the corresponding explanation to the text (lines 214-225).

Finally, several aspects of the cryo-EM processing and data analysis require clarification. These include corrections to models and tables (for example, Table 2 reports 1.82% Ramachandran

outliers in one model, which is unexpected at the reported resolution and should be revised). In addition, local-resolution maps appear to show unusual features (e.g., the TMD map in Extended Data Fig. 1, or the first LBD map in Extended Data Fig. 2, where the map seem to display substantially higher resolution than stated considering FSC graph). The authors should carefully re-examine these maps and provide a clearer description of “composite” and consensus maps, including how composite-map resolution was estimated and why particle numbers differ between open-state consensus, composite, and focused maps. No Ramachandran outliers are expected.

We have corrected our structural models to reduce the percentage of Ramachandran outliers (**Supplementary Table 1**). We have also examined all illustrated maps in the supplementary figures and made sure that they are all correct (see also a response to the other comment for **Extended Data Fig. 1** below). We also added a more detailed description of the composite and consensus maps, and how they were made (lines 494-497).

Other comments:

- Line 65 “For instance, GluA2-containing AMPARs are generally Ca²⁺-impermeable due to RNA editing that substitutes 65 glutamine (Q) for arginine (R) at position 586, the Q/R site, within the channel pore.

The distinction between Ca²⁺-permeable and Ca²⁺-impermeable AMPARs should be revisited in light of recent publications (Miguez-Cabello et al., Nature, 2025).

The text has been corrected accordingly (line 65).

-Line 74 “all available structures of the homomeric core alone have suffered from both low resolution and absence of the critical re-entrant M2 loop of the transmembrane domain (TMD), which lines the pore in conjunction with helix M3 and forms the channel’s selectivity filter”

This statement is not accurate. The structure of wild-type GluA4 without auxiliary proteins was recently determined at side-chain resolution and showed clear density for the M2 region (Vega-Gutierrez et al., NSMB, 2025).

Our manuscript had been submitted before the mentioned paper was released. We have corrected the text accordingly and added the mentioned citation (line 74).

-Line 151: “Due to the flexibility of the ATD layer that worsens the overall resolution of the full length receptor map and as it does not contribute to AMPAR functionalities, we focused our structural analysis on the LBD-TMD region”.

This assertion is not precise. Multiple recent studies (Ivica et al., NSMB 2024; Larsen et al., JBC 2024; Zhang et al., Nature 2023) demonstrate that the ATD (specifically the GluA2 ATD tetramer interface), plays a critical role in recovery from desensitization. While it is true that the ATD is flexible, current cryo-EM methods can reliably capture this variability, especially in GluA2-containing receptors, where the ATD dimer interface is stable. Furthermore, having the ATD structure is essential to correctly assign subunits within the tetramer. Given the high sequence conservation in the LBD and TMD, the manuscript does not presently provide sufficient evidence for the subunit positioning reported. Therefore, the analysis of the NTD layer should also be presented. Have the authors identified unambiguous, subunit-specific residues in the density maps corresponding to GluA2 or GluA1?

We have obtained ATD reconstruction for GluA1/A2Glu+RR2b. Using densities for the ATD layer, we have also included illustrations of cryo-EM density for the carbohydrates at the predicted ATD sites that uniquely distinguish subunits GluA1 and GluA2 and unambiguously verify the established order of subunits in the GluA1/A2 tetramer (**Supplementary Fig. 4g-j**), consistent with the previous reports^{3,20,22}.

- Figure 6:

Panel (e) contains incorrect references: reference 65 is a proteomic study and does not report GluA1/A2/TARP8 functional characterization. Additionally, comparing recovery data across whole-cell and outside-out patch recordings is problematic, as major differences in desensitization (both entry and recovery) have been shown in back-to-back experiments (Zhang et al., Nature, 2023). The table also mixes data from constructs with shortened linkers and constructs with wild-type linkers, even though linker length is known to affect recovery kinetics in AMPAR–TARP complexes. Considering this concerns as well as the fact that most data in panel (e) have not been generated in this manuscript, it may be better to exclude it from a main text figure.

Reference 65 (now reference 76) is for the study of Schwenk et al. (Neuron, 2012)²⁷, which indeed focuses on AMPAR proteomics. However, this same study also reports electrophysiological recordings using giant outside-out patches excised from *Xenopus* oocytes (see Figure 5). The results of these experiments are the ones that have been referenced in **Fig. 6e**. Additionally, if the application system has appropriate kinetic characteristics (fast enough speed of solution exchange), whole-cell and outside-out recordings provide exactly same values of kinetic parameters, which we confirmed in our experiments numerous times. In our hands, the kinetics of AMPA receptor currents measured in the whole-cell and outside-out modes are identical. Nevertheless, as requested, panel e has been moved to **Supplementary Fig. 9** and maximum possible information about the corresponding experiments has been added to the figure legend.

- Extended Data Figure 1:

Please check the handedness of the grey map in the classification step, it looks like it is inverted. Also Re-examine the local-resolution calculations for the TMD map; the left side displays an unusual blue pattern that should be verified. Please complete the figure legend.

The handedness of maps in the middle of the processing pipeline, such as during particle cleanup using heterogenous refinement, has no influence on the final reconstructions. Nevertheless, we have replaced the previous illustration with the corresponding right-handed one (see new **Supplementary Fig. 1**). The “unusual” blue pattern reflects the higher resolution of the TMD core (pore region) when the refinement is focused on the TMD dimer (not monomer!). Additional information has been added to the figure legend (lines 1042 - 1054).

-Cryo-EM analysis of the open state:

Please, describe with a figure the 3D processing strategy for the open state. Currently, the number of particles in Table 1 and 2 do not match the processing strategy defined in the text (131,148 particles vs 151,693 -LBD, 74183 for the TMD and 61870 particles): has the composite map been obtained with two different sets of particles? Please clarify this. In general Table 1 and Table 2 and methods are not very clear: for example, how is the resolution of the composite map calculated? For the CNIH complex, both composite and consensus maps are reported, but the focused maps used to build the composite map are not documented. Composite maps should be reserved for visualization only, particularly when derived from different particle subsets.

As requested, we have added **Supplementary Figs. 1-3** describing the processing workflow and added a more detailed description of the processing strategy (lines 494-497). Of course, the LBDs and TMDs of CNIH and non-CNIH bound complexes have different numbers of particles, because they were obtained from different subpopulations of particles. And the differences in numbers are based on further classifications/cleanup of particles from the consensus, not completely different particles altogether. With respect to the resolution of the composite maps, we followed the recommendations of the EMDB. According to EMDB website: “EMDB recommends using the mean resolution of the focused maps as the resolution of the composite map”.

-Cryo-EM density details:

Please include CNIH2 density in Extended Data Figure 4, especially for regions that distinguish CNIH2 from CNIH1. Also provide density features confirming GluA1 vs. GluA2 subunit identity (e.g., the Q/R site, R/G site, or other subunit-specific residues that support the assignment).

For CNIHs, we made a separate **Supplementary Fig. 6**, where we illustrate the similarity of the fit of all four CNIHs, CNIH1-4, into the density (panels a-e) as well as give examples that

distinguish fit of different CNIH fragments into the density (panels f-t). In addition, we have also included illustrations of cryo-EM density for the carbohydrates at the predicted ATD sites that uniquely distinguish subunits GluA1 and GluA2 and unambiguously verify the established order of subunits in the GluA1/A2 tetramer (**Supplementary Fig. 4g-j**), consistent with the previous reports^{3,20,22}.

-Extended Data Fig. 7:

Correct the reference for the GluA1/A2/TARP8 dataset and clearly specify differences in recording configuration (whole-cell vs. outside-out) and in constructs used (wild type vs. modified).

Done. The reference for GluA1/A2-y8 was left as is²⁷ as it does indeed report the corresponding electrophysiology recordings from outside-out patches from xenopus oocytes (see supplemental materials). We have also included the information about the constructs and recording configuration into the **Supplementary Fig. 9g-i** legend (lines 1158 – 1170).

We thank Reviewers for their time and effort in reviewing our manuscript. We have made additional changes that are outlined in our responses below.

Reviewer #1 (Remarks to the Author):

The authors have addressed my concerns and made extensive revisions to their manuscript in response to all three of the initial reviews. The important advances in this study were noted in the initial reviews and the revised manuscript is much improved. No additional concerns were noted.

We thank Reviewer #1 for the positive and generous assessment of our work.

Reviewer #2 (Remarks to the Author):

The authors have addressed my concerns.

Reviewer #3 (Remarks to the Author):

Thank you very much for the revised version of the manuscript.

The authors have addressed most of my comments. I still think that the structural work presented here would be more relevant if WT constructs were used, as current GluA1/A2 structures in complex with auxiliary proteins are WT. Still, the manuscript presents very useful data, including CNIH1 binding and open states without auxiliary proteins. The MS data, as well as the details of the cryo-EM maps in the CNIH TMD, clearly support the binding of CNIH1 to AMPARs in HEK cells. This opens new questions about a possible role of CNIH1 in trafficking or kinetic modulation.

We thank Reviewer #3 for the generous assessment of our work.

Please add the mass spectrometry methods.

The mass spectrometry methods have been added to the manuscript.

I do not have any more comments.